**Microbial dormancy and its impacts on Arctic terrestrial ecosystem carbon budget**

Junrong Zha and Qianlai Zhuang

Department of Earth, Atmospheric, and Planetary Sciences and Department of Agronomy, Purdue University, West Lafayette, IN 47907 USA

Submitted to: *Biogeoscience*

Correspondence to: qzhuang@purdue.edu

**Abstract**
**A large amount of soil carbon in the Arctic terrestrial ecosystems could be emitted as**
**greenhouse gases in a warming future. However, lacking detailed microbial processes such**
**as microbial dormancy in current biogeochemistry models might have biased the**
**quantification of the regional carbon dynamics. Here the effect of microbial dormancy was**
**incorporated into a biogeochemistry model to improve the quantification for the last and**
**this century. Compared with the previous model without considering the microbial**
**dormancy, the new model estimated the regional soils stored 75.9 Pg more C in the**
**terrestrial ecosystems during the last century, and will store 50.4 Pg and 125.2 Pg more C**
**under the RCP 8.5 and RCP 2.6 scenarios, respectively, in this century. This study**
**highlights the importance of the representation of microbial dormancy in earth system**
**models to adequately quantify the carbon dynamics in the Arctic.**






**1. Introduction**

The land ecosystems in northern high latitudes (>45 ºN) occupy 22% of the global surface and store over 40% of the global soil organic carbon (SOC) (McGuire & Hobbie, 1997; Melillo et al., 1993; Tarnocai et al., 2009; Hugelius et al., 2014). During the past decades, a greening accompanying a warming in the region has been documented (Zhou et al., 2001; Lloyd et al., 2002; Stow et al., 2004; Callaghan et al., 2005; Tape et al., 2006). The regional carbon dynamics are expected to loom large in the global carbon cycle and exert large feedbacks to the global climate system (McGuire et al., 2009; Davidson & Janssens, 2006; Bond-Lamberty & Thomson, 2010).

To date, numerous ecosystem models have been developed to project the feedbacks between terrestrial ecosystem carbon cycling and climate (Raich et al., 1991; Zhuang et al., 2001, 2002, 2015; Parton et al., 1993; Knorr et al., 2005; Running & Coughlan, 1988), but they can bias their quantifications due to missing detailed microbial mechanisms in these models (Schmidt et al., 2011; Todd-Brown et al., 2013; Conant et al., 2011; Treseder et al., 2011). Microorganisms play a central role in decomposition of litter and soil organic carbon, which further governs the global carbon cycling and climate change (Xu et al., 2014; Treseder et al., 2011; Wang et al., 2015). An emerging field of research has begun to incorporate microbial ecology into existing process-based models to represent decomposition in ways that include important microbial processes that were previously ignored (Zha & Zhuang, 2018; Schimel & Weintraub, 2003; Allison et al., 2010; German et al., 2012). These microbial-based models tend to better reproduce field and satellite observations than traditional ones that treat soil decomposition as a first-order decay process without considering microbial activities (Treseder et al., 2011; Wieder et al., 2013; Todd-Brown et al., 2011; Lawrence et al., 2009; Moorhead et

al., 2006). However, some vital microbial traits such as microbial dormancy and community
shifts are still rarely explicitly considered in large-scale ecosystem models (Wieder et al., 2015),
and this may introduce notable uncertainties (Graham et al., 2014, 2016; Wang et al., 2015;
Bouskill et al., 2012; Kaiser et al., 2014).

Dormancy is broadly recognized as a strategy for microorganisms to cope with periodical

environmental stresses (Harder & Dijkhuizen, 1983). When environmental conditions are
unfavorable for growth, microbes switch to a dormant state, which is a reversible state of low to
zero metabolic activity (Stolpovsky et al., 2011; Lennon & Jones, 2011). In this state,
biogeochemical processes such as soil decomposition are slow (Blagodatskaya et al., 2013).  At
any given time, there is only a fraction of, likely below 50%, metabolically active microbes  in
natural soils (Wang et al., 2015; Stolpovsky et al., 2011). Soil decomposition and nutrient
cycling mainly depend on these active microbes because only active ones can consume organic
matter and replicate themselves (Wang et al., 2015; Blagodatskaya et al., 2014).  To date, most
existing biogeochemistry models use total rather than active microbial biomass as an indicator of
microbial activities (Wieder et al., 2015), which could bias the estimates of soil decomposition
and ecosystem carbon budget (Hagerty et al., 2014; He et al., 2015). Especially, the Arctic
terrestrial ecosystems are nitrogen-limited, neglecting microbial dormancy will lead to incorrect
estimates of nitrogen availability through soil decomposition, failing to capture nitrogen
feedbacks to carbon dynamics (Wang et al., 2015; Stolpovsky et al., 2011; Thullner et al., 2005).
Furthermore, the Arctic has experienced a marked seasonality of active and dormant microbial
cycles and the above-global-average warming, which might have increased the proportion of
active microbes in soils (He et al., 2015). Thus, incorporating dormancy effects will improve
model realism to provide a better projection of the Arctic carbon dynamics.
This study incorporated the effects of microbial dormancy trait into an extant process-
based biogeochemistry model (MIC-TEM) (Zha & Zhuang, 2018; He et al., 2015). The dormant
and active microbial physiology has been considered explicitly in the new version of model
(MIC-TEM-dormancy).  The revised model was parameterized, validated, and then applied to
evaluate the carbon dynamics during the last and this centuries in the Arctic terrestrial
ecosystems (north 45 ºN above). By comparing the results of MIC-TEM-dormancy and MIC-
TEM, we can show that incorporating microbial dormancy may produce a much different
prediction in historical and future carbon budget.

**2. Methods**
**2.1 Overview**
Due to the importance of microbial dormancy, some recent work has been done to consider
the metabolic activation and deactivation of microbes in soil and its effects on soil carbon (C)
dynamics and climate feedbacks. For example, Wang et al. (2015) has incorporated transformation
processes between active and dormant states to develop two versions of MEND, that is, MEND
with and without dormancy. The two versions of the model have been applied to quantify the
carbon decomposition in laboratory incubations of four soils. Salazar et al. (2018) have also taken
microbial dormancy into account to compare their predictions of microbial biomass and soil
heterotrophic respiration ($R_H$) under simulated cycles of stressful (dryness) and favorable (wet
pulses) conditions.  Our study extends those modeling studies to the whole Arctic region by
developing a more detailed biogeochemistry model considering the dormancy impacts.  Below, we
first describe how we developed the new model (MIC-TEM-dormancy) by incorporating the
microbial dormancy trait into an existing microbial-based biogeochemistry model (MIC-TEM).
Second, we discuss how parameterization and validation of MIC-TEM-dormancy model were
conducted using observed net ecosystem exchange data, and heterotrophic respiration data at
representative sites. Third, we presented how the model was applied to northern high latitudes
(above 45 °N) for the 20th and 21st centuries and discussed the dormancy effects on regional carbon
budget.

**2.2 Model description**
A non-dormancy version of biogeochemistry model (MIC-TEM) has been developed by
incorporating a microbial module (Allison et al., 2010) into an extant large-scale biogeochemical
model (TEM) to explicitly (Zhuang et al., 2003) consider the effects of microbial dynamics and
enzyme kinetics on carbon dynamics (Zha & Zhuang, 2018). Here we further advanced the MIC-
TEM by incorporating algorithms that describe the effects of microbial dormancy dynamics
based on He et al. (2015). Different from He et al. (2015), in which microbial module was driven
with existing data of carbon stocks and fluxes, our study incorporated the microbial module into
an extant MIC-TEM that simulates carbon data dynamically. This coupling enables us to
extrapolate our model to whole northern high-latitudes region, rather than only for temperate
forest region in He et al. (2015).  In our new model (MIC-TEM-dormancy), microbial biomass
pool was divided into two fractions, including the dormant and active microbial biomass pools.
The two microbial biomass pools and the reversible transition between them have been
considered explicitly in the new model (Figure 1), which was ignored in MIC-TEM.
In previous MIC-TEM, heterotrophic respiration ($R_H$) is calculated as:
$$R_H = ASSIM \times (1\text{-}CUE) \qquad (1)$$
Where ASSIM and CUE represent microbial assimilation and carbon use efficiency, respectively.
For detailed carbon dynamics in MIC-TEM, see Zha & Zhuang (2018).

Here we revised MIC-TEM by incorporating microbial dormancy dynamics according to

He et al. (2015). In MIC-TEM-dormancy, the soil heterotrophic respiration $R_H$ is comprised of
three parts: the maintenance respiration from the active and dormant microorganisms and the $CO_2$
production through the process of microbial assimilation (He et al., 2015):
$$R_H = m_R Q_{10mic}^{\frac{temp-15}{10}} B_a + \beta m_R Q_{10mic}^{\frac{temp-15}{10}} B_d + CO_2 \qquad (2)$$
where the first two terms are maintenance respiration from the active and dormant
microorganisms, respectively. The last term is the $CO_2$ produced during the process of microbial
assimilation.
For first two terms, $B_a$ and $B_d$ represents the active and dormant microbial biomass pool,
respectively. The parameter $m_R$ denotes the specific maintenance rate at active state ($h^{-1}$), and $\beta$
is the ratio of dormant maintenance rate to active maintenance rate. Thus, $\beta m_R$ denotes the
maximum specific maintenance rate at dormant state. Temperature sensitivity was expressed as
the $Q_{10}$ function ($Q_{10}^{\frac{temp-15}{10}}$), where temp is soil temperature at top 20 cm (units: °C).
For the third term, the $CO_2$ produced through microbial assimilation is calculated as in He et al.
(2015) and Allison et al. (2010):
$$CO_2 = ASSIM \times (1 - Y_g) \qquad (3)$$
Where ASSIM represents the microbial assimilation and the parameter $Y_g$ represents carbon use
efficiency.  Microbial assimilation (ASSIM) is calculated as in He et al. (2015):
$$ASSIM = \frac{1}{Y_g} \frac{\Phi}{\alpha} m_R Q_{10enz}^{\frac{temp-15}{10}} Ba \left(\frac{CN_{soil}}{CN_{mic}}\right)^{0.6} \qquad (4)$$
Here parameter $\alpha$ is maintenance weight ($h^{-1}$), $CN_{soil}$ and $CN_{mic}$ denotes the C:N ratios of soil and
that of microbial biomass. Besides, $\Phi$ is the substrate saturation level and defined as in He et al.
(2015) and Wang et al. (2014):

$\Phi = \dfrac{S}{K_S + S}$              (5)

Where $K_s$ is the half saturation constant for substrate uptake as indicated by the Michaelis–Menten
kinetic, and S is soluble C substrates that are directly accessible for microbial assimilation (Wang
et al., 2014). Here we quantified concentration of soluble C substrates that are directly accessible
for microbial assimilation by using conceptual framework from Davidson et al. (2012):

$S = \text{Soluble } C \times D_{liq} \times \theta^3$         (6)

The term 'Soluble C' denotes the state variable of soluble carbon pool. $D_{liq}$ is the diffusion
coefficient of the substrate in the liquid phase, and is formulated as:

$D_{liq} = 1/(1\text{-}BD/PD)^3$   (7)

Where BD is the bulk density and PD is the soil particle density. $\theta$ is the volumetric soil moisture.
Different from MIC-TEM, the transitions between active and dormant microbial biomass are
included in MIC-TEM-dormancy.

$B_{a \to d} = (1 - \Phi) m_R Q_{10mic}^{\frac{temp-15}{10}} B_a$       (8)

$B_{d \to a} = \Phi m_R Q_{10mic}^{\frac{temp-15}{10}} B_d$        (9)

Where $B_{a \to d}$ and $B_{d \to a}$ denote the transition from the active to dormant microbe and from the
dormant to active microbe, respectively (He et al., 2015; Wang et al., 2014). Thus, dormancy rate
is affected by active and dormant biomass, soil temperature (temp) and soil moisture ($\theta$ in $\Phi$).
The active microbial biomass ($B_a$) is modeled as (He et al., 2015; Wang et al., 2014):
$\dfrac{dBa}{dt} = ASSIM \times Y_g - m_R Q_{10mic}^{\frac{temp-15}{10}} B_a - B_{a \to d} + B_{d \to a} - DEATH - EPROD$      (10)

Where DEATH and EPROD denotes microbial biomass death and enzyme production, which are modeled as proportional to active microbial biomass with constant rates $r_{death}$ and $r_{EnzProd}$ (Allison et al., 2010):

$$DEATH = r_{death} \times Ba \qquad (11)$$

$$EPROD = r_{EnzProd} \times Ba \qquad (12)$$

Where $r_{death}$ and $r_{EnzProd}$ are the rate constants of microbial death and enzyme production, respectively.

The dormant microbial biomass ($B_d$) is modeled as (He et al., 2015; Wang et al., 2014):

$$\frac{dB_d}{dt} = -\beta m_R Q_{10mic}^{\frac{temp-15}{10}} B_d + B_{a \to d} - B_{d \to a} \qquad (13)$$

The Soluble C pool is modeled as (He et al., 2015; Allison et al., 2010):

$$\frac{d\,Soluble\,C}{dt} = DECAY - ASSIM + ELOSS + DEATH \qquad (14)$$

Where DECAY represents the enzymatic decay of soil organic carbon (SOC), and ELOSS represents the loss of enzyme.

DECAY is regulated by enzyme biomass (ENZ), soil organic carbon (SOC), soil temperature, and substrate quality (He et al., 2015):

$$DECAY = V_{max} \times Q_{10enz}^{\frac{temp-15}{10}} \times ENZ \times \frac{SOC}{Km_{uptake} + SOC} \times (120 - CN_{soil}) \qquad (15)$$

Where $V_{max}$ is the maximum SOC decay rate, $Km_{uptake}$ is half saturation constant for enzymatic decay.

ELOSS is modeled as a first-order process (Allison et al., 2010) to represent enzyme turnover:

$$ELOSS = r_{enzloss} \times ENZ \qquad (16)$$

Where $r_{enzloss}$ is the rate constant of enzyme loss.

The soil organic carbon pool (SOC) is modeled as:

$$\frac{dSOC}{dt} = \text{Litterfall} - \text{DECAY} \qquad (17)$$

Where Litterfall is estimated as a function of vegetation carbon (Zhuang et al., 2010).
Last, enzyme pool (ENZ) is modeled as:
$$\frac{dENZ}{dt} = \text{EPROD-ELOSS} \qquad (18)$$

With the modification of microbial carbon dynamics by considering microbial life-history trait,
soil decomposition is changed since it is controlled by microbes. When microbial dormancy is
considered, the number of active microbes that participate in soil decomposition is much less. The
changes in soil decomposition directly influence the amount of soil respiration, and further
influence soil nitrogen (N) mineralization that determines soil N availability for plants, affecting
gross primary production (GPP). Since both GPP and $R_H$ can be affected by microbial dormancy,
net ecosystem production (NEP) will also be affected.

**2.3 Model parameterization and validation**

The detailed description of parameters that are related to microbial dormancy can be found

in He et al. (2015) (Table 1). Here we calibrated the MIC-TEM-dormancy at six representative
sites with gap-filled monthly net ecosystem productivity (NEP, $gCm^{-2}mon^{-1}$) data in northern high
latitudes (Table 2). Site-level climatic data and soil texture data were organized for driving model.
All sites information can be found on AmeriFlux network (Davidson et al., 2000). The results for
model parameterization were presented in Figure 2. We conducted the parameterization using a
global optimization algorithm known as SCE-UA (Shuffled complex evolution) method (Duan et
al., 1994). An ensemble of 50 independent sets of parameters were performed based on prior ranges
from literature (Table 1) to minimize the difference between the monthly simulated and measured
NEP at the chosen sites. The cost function of the minimization is:
$$\qquad \text{Obj} = \sum_{i=1}^{k}(\text{NEP}_{obs,i} - \text{NEP}_{sim,i})^2 \qquad \qquad (17)$$
Where $\text{NEP}_{obs,i}$ and $\text{NEP}_{sim,i}$ are the observed and simulated NEP, respectively. k is the number of
data pairs for comparison. Except for the parameters of microbial dormancy, other parameters are
derived directly from MIC-TEM (Zha & Zhuang, 2018). The optimized parameters were used for
model validation and regional simulations.
For model validation, we chose another six sites that containing monthly NEP data from
AmeriFlux network (Table 3). Moreover, we also conducted site-level validations with monthly
soil respiration data from AmeriFlux network and Fluxnet dataset. The site information was
provided in Table 4. For these sites, we assumed 50% of soil respiration was heterotrophic
respiration ($R_H$) for forest (Hanson et al., 2000), 60% and 70% of that was $R_H$ for grassland (Wang
et al., 2009) and tundra (Billings et al., 1977). Because there is a limited amount of available RH
data, we could not conduct a regional validation for all pixels in northern high latitudes. Instead,
we extracted 61 sites providing data of average annual heterotrophic respiration from ORNL global
Soil Respiration Dataset (https://daac.ornl.gov/SOILS/guides/SRDB_V4.html, Bond-Lamberty et
al., 2018) for model validation. The site-level observed average annual $R_H$ was used to compare
with simulated annual $R_H$ by MIC-TEM-dormancy and MIC-TEM. The MIC-TEM-dormancy was
run at monthly time step to keep consistent with the time step of MIC-TEM. Although microbial
dynamics occur at fine temporal scales (Tang & Riley, 2014), we can still quantify the cumulative
impacts of microbial dynamics on carbon and nitrogen cycling at monthly time by not changing
the model structure.

**2.4 Spatial extrapolation**

For historical simulations during the 20[th] century, two sets of regional simulations using

MIC-TEM-dormancy and MIC-TEM at a spatial resolution of 0.5° latitude × 0.5° longitude were
conducted. Our model simulation contains two parts: spin-up and transient simulation. A typical
spin-up was conducted to get the model to a steady state for each spatial location, which will be
used as initial conditions for transient simulations (McGuire et al., 1992). During spin-up
procedure, cyclic forcing data was used to force the model run, and repeated continuously until
dynamic equilibrium was achieved at which the modeled state variables show a cyclic pattern or
become constant.  Specifically, this study used the monthly historical climate data from 1900 to
1940 to repeatedly drive the model for the spin-up. Before spin-up procedure, the model was
initialized with default built-in carbon stocks (Raich et al., 1991). During transient simulations,
the calibrated ecosystem-specific parameters were used for regional simulations. The previous
dynamic equilibrium was used as initial value for transient simulation. The historical climatic
forcing data, including the monthly air temperature, precipitation, cloudiness, and atmospheric
$CO_2$ concentrations, were organized from the Climatic Research Unit (CRU TS3.1) from the
University of East Anglia (Harris et al., 2014). We also used gridded data of soil texture (Zhuang
et al., 2003), elevation (Zhuang et al., 2015), and potential natural vegetation (Melillo et al., 1993)
from literatures. In our model, we assumed that soil texture, elevation, and potential natural
vegetation data only vary spatially, not vary over time (Zhuang et al., 2015).

In addition, regional simulations over the 21[st] century were conducted under two

Intergovernmental Panel on Climate Change (IPCC) climate scenarios (RCP 2.6 and RCP 8.5).
The future climatic forcing data under these two climate change scenarios were derived from the
HadGEM2-ESmodel,   which   is   a   member   of   CMIP5project213   (https://esgf-
node.llnl.gov/search/cmip5/). Then the regional estimations were obtained by summing up the
gridded outputs for our study region. The positive simulated NEP represents a $CO_2$ sink from the
atmosphere to terrestrial ecosystems, while a negative value represents a source of $CO_2$ from
terrestrial ecosystems to the atmosphere.
**2.5 Parameter equifinality effects**
Our previous studies using TEM has demonstrated that equifinality derived from site-level
parameterization will affect the uncertainty in the estimation of regional carbon dynamics (Tang
and Zhuang, 2008, 2009). Here equifinality refers to that a number of sets of parameters result in
model simulations that all match the data similarly well. To quantify this effect on our simulation
uncertainty, we conducted ensemble regional simulations with 50 sets of parameters for both
historical and future studies. The 50 sets of parameters were obtained according to the method in
Tang and Zhuang (2008).
**3. Results**
**3.1 Inversed Model Parameters and model validation**
Using SCE-UA ensemble method, 50 independent sets of parameters were converged to
minimize the objective function. Then the optimized parameters are calculated as the mean of these
50 sets of inversed parameters. The boxplot of parameter posterior distributions reflects different
ecosystem properties at these sites (Figure 3). For instance, growth yield was higher in tundra types
than in forests, meaning microorganisms in environment with higher energy limitation tend to
enhance the efficiency of energy transportation. Besides, alpha, the maintenance weight, was also
higher in tundra types than in forests. From the plot for parameter beta, the ratio of dormant
maintenance rate to specific maintenance rate for active biomass in tundra types is lower than that
in forest types. Other microbial related parameters did not differentiate much among different
vegetation types.
After parameterization, the MIC-TEM-dormancy was validated with monthly NEP data for
six representative ecosystems, and the comparisons between monthly observed NEP and
simulated NEP were presented in Figure 4. With the optimized parameters, the dormancy-based
model was used to reproduce NEP to compare with the measured NEP (Table 5). The $R^2$ ranges
from 0.67 for Atqasuk to 0.93 for Bartlett Experimental Forest (Table 5). Generally, our new
model performs better for forest ecosystems than for tundra ecosystems. Compared with MIC-
TEM, dormancy model performs better for alpine tundra, temperate coniferous forest, and
grassland. For other sites, both models show similar performance (Table 5).   Besides, a set of
monthly soil respiration data were selected to evaluate the estimated $R_H$.  The comparisons
between monthly observed $R_H$ and simulated $R_H$ from two contrasting models were conducted
(Figure 5). MIC-TEM-dormancy has higher $R^2$ and lower root mean square error (RMSE) (Table
6).  Sixty-one sites with average annual $R_H$ in northern high-latitude regions were used to further
evaluate the new model performance. The dormancy model has lower intercept and slope with $R^2$
of 0.45, while $R^2$ of MIC-TEM is 0.3 (Figure 6). These analyses indicate that new model is more
realistic in representing $R_H$ by considering microbial dormancy. This difference in $R_H$ further
affects soil available nitrogen dynamics, influencing nitrogen uptake by plants, the rate of
photosynthesis and NPP (Zhuang et al., 2015; Zha et al., 2018; Thullner et al., 2005).

**3.2 Regional carbon dynamics during the 20th century**
Regional extrapolation with both models estimated a regional carbon sink but with different
magnitudes (Figure 7c). With optimized parameters, MIC-TEM estimated a regional carbon sink
of 77.6 Pg with the interannual standard deviation of 0.21 Pg C yr$^{-1}$ during the 20th century.
However, MIC-TEM-dormancy nearly doubles the sink at 153.5 Pg with the interannual standard
deviation of 0.12 Pg C yr$^{-1}$ during the last century (Figure 7c). At the end of the century, MIC-
TEM estimated that NEP reaches 1.0 Pg C yr$^{-1}$ in comparison with MIC-TEM-dormancy estimates
of 1.5 Pg C yr$^{-1}$ (Figure 7c). Both models simulated similar trends for regional NPP, $R_H$ and NEP
(Figure 7). Generally, they show an increasing trend in the 20th century (Figure 7). Meanwhile,
with optimized parameters, MIC-TEM-dormancy estimated NPP and $R_H$ at 7.94 Pg C yr$^{-1}$ and 6.4
Pg C yr$^{-1}$, which are 5.8% and 16.3% less than the estimations from MIC-TEM, respectively
(Figures 7a and 7b). This pronounced difference of NEP between two models comes from the
disparity between the simulated NPP and $R_H$ with them since NEP is calculated as the difference
between NPP and $R_H$. Without considering dormancy, MIC-TEM estimates more active microbial
biomass since it assumes the whole microbial biomass pool will participate in soil decomposition.
The fact is only active part of microbial biomass can affect organic matter decomposition, meaning
MIC-TEM overestimates $R_H$. On the other hand, overestimation of $R_H$ can induce higher nitrogen
uptake by plants, which will accelerate rate of photosynthesis and further enhance NPP projection.
Although MIC-TEM estimates higher NPP and $R_H$ than MIC-TEM-dormancy does, NEP estimated
from MIC-TEM is actually lower.

The average annual seasonal patterns of NPP, $R_H$ and NEP during the 1990s were also

organized from regional simulations with two models (Figure 8). Temporally, both models
projected higher NPP and $R_H$ in summer than in winter (Figures 8a and 8b) due to higher soil
temperature and moisture (McGuire et al., 1992). Setting the $R_H$ projection from MIC-TEM as a
baseline, MIC-TEM-dormancy averagely projected 33% less $R_H$ in summer (May to September),
and 30% more in winter (other months) (Figure 8b), which indicates that without dormancy,
model tends to estimate lower soil respiration compared to dormancy model due to ignorance of
dormant respiration in winter but estimate higher soil respiration due to higher estimation of
active biomass in summer. In the meantime, seasonal cycle of NPP with MIC-TEM-dormancy
shows a relative flattening pattern compared with MIC-TEM, which is similar to seasonal cycle
of $R_H$ (Figure 8a).  Though $R_H$ and NPP show the similar seasonal patterns, NEP can still show
different pattern. Here seasonal cycles of NEP with models are close to each other (Figure 8c),
but dormancy model projected slightly higher NEP in summer.
**3.3 Regional carbon dynamics during the 21$^{st}$ century**
Under the RCP 8.5 scenario, both models estimated the region acts as a carbon sink (Figure
9). The MIC-TEM-dormancy predicted a C accumulation of 129.9 Pg by the end of this century.
with the interannual standard deviation of 0.13 Pg C yr$^{-1}$, whereas MIC-TEM estimates a C
accumulation of 79.5 Pg with the interannual standard deviation of 0.37 Pg C yr$^{-1}$ during the 21$^{st}$
century (Figure 9). Thus, MIC-TEM-dormancy estimates an increase of 50.4 Pg regional carbon
sequestration relative to MIC-TEM, with less interannual variation (Figure 9). Under this
scenario, both models predict similar temporal trends for NEP, namely increasing from the 2000s
and then decreasing from the 2070s onward (Figure 9). MIC-TEM-dormancy predicts that
carbon sink reaches 1.36 Pg C yr$^{-1}$ in the 2090s, which is 0.26 Pg C yr$^{-1}$ more than projection of
MIC-TEM. Moreover, MIC-TEM-dormancy estimated NPP and $R_H$ at 10.2 Pg C yr$^{-1}$ and 8.9 Pg
C yr$^{-1}$, which are 1.3 Pg C yr$^{-1}$ and 1.8 Pg C yr$^{-1}$ less than the estimations from MIC-TEM,
respectively (Figure 9).
Under the RCP 2.6 scenario, the cumulative NEP from two models diverged by 125.2 Pg C
by 2100. The trajectory of inter-annual NEP estimated with the two models also diverged. The
MIC-TEM predicted the region fluctuates between carbon sinks and sources, and totally acts as a
carbon source of 1.6 Pg C with the interannual standard deviation of 0.24 Pg C yr$^{-1}$ during the
21$^{st}$ century. In contrast, MIC-TEM-dormancy projected the region acts as a carbon sink of 123.6
Pg C with an interannual standard deviation of 0.1 Pg C yr$^{-1}$ (Figure 9). MIC-TEM-dormancy
estimates NPP and $R_H$ at 9.9 Pg C yr$^{-1}$ and 8.7 Pg C yr$^{-1}$, which are 0.5 Pg C yr$^{-1}$ and 1.7 Pg C yr$^{-1}$
less than the estimations from MIC-TEM, respectively (Figure 9). Moreover, simulations under
the two contrasting climate scenarios (RCP 2.6 and RCP 8.5) exhibit a large difference of 81.1
Pg C of cumulative NEP during the 21$^{st}$ century by MIC-TEM, but only 6.3 Pg C of that by
MIC-TEM-dormancy. This difference indicates microbes provide a resistant response to climate
change due to dormancy to some extent (Treseder et al., 2011).

The average annual seasonal patterns of NPP, $R_H$ and NEP during the 2990s by two

models were also presented (Figure 10). MIC-TEM-dormancy estimated higher $R_H$ in winter, but
lower $R_H$ in summer under both future scenarios (Figure 10). NPP is the same in winter with or
without dormancy, and in the late summer is higher than that without dormancy, especially in the
RCP 8.5 scenario.  The combined flattening patterns of NPP and $R_H$ result in different patterns
for NEP. Under the RCP 2.6 scenario, MIC-TEM-dormancy predicts higher NEP from June to
October, but lower NEP from January to April compared to MIC-TEM (Figure 10). Under the
RCP 8.5 scenario, MIC-TEM-dormancy predicts higher NEP from June to September, but much
lower NEP in other months than MIC-TEM (Figure 10).
**3.4 Regional uncertainty considering equifinality effects during 20$^{th}$ and 21$^{st}$ centuries**

The ensemble simulations for the 20$^{th}$ century is shown in Figure 11. Given the

uncertainty in parameters, MIC-TEM-dormancy predicted that the regional cumulative carbon
ranges from a carbon loss of 28.2 Pg to a carbon sink of 362.1 Pg by different ensemble
members, with a mean of 71.2±54.8 Pg (Figure 11). For the 21$^{st}$ century, MIC-TEM-dormancy
predicted that the region acts from a carbon source of 49.3 Pg C to a carbon sink of 296.5 Pg C,
with a mean of 112.7±116.5 Pg under the RCP 2.6 scenario (Figure 12). Under the RCP 8.5
scenario, MIC-TEM-dormancy predicted that the region acts from a carbon source of 27.1 Pg C
to a carbon sink of 401.3 Pg C, with a mean of 143.1±162.5 Pg (Figure 12).

**4. Discussion**

Soils are the largest carbon repository in the terrestrial biosphere and hold 2.5 times more
carbon than the atmosphere (Frey et al., 2013; Schlesinger & Andrews, 2000). Especially, a
significant portion of soil organic carbon stored in northern high latitudes (Tarnocai et al., 2009).
Besides, the magnitude of the warming in these regions is larger, almost twice, that of the global
average (Serreze & Francis, 2006) and the changing climate is expected to alter the carbon cycle
through influencing the activities of microorganisms in controlling soil decomposition (Manzoni
et al., 2012; Melillo et al., 2011). Therefore, explicit consideration of microbial traits and
functions in large-scale biogeochemistry models is necessary for better quantification of carbon-
climate feedbacks (Thullner et al., 2005; Wang et al., 2015). Our regional simulations with two
contrasting models (MIC-TEM, MIC-TEM-dormancy) indicate the region was a carbon sink in
past decades, which is consistent with results from other process-based models (White et al.,
2000; Houghton et al., 2007; McGuire et al., 2009; Schimel, 2013). However, the magnitudes of
this sink are quite different in two models. Moreover, MIC-TEM-dormancy predicts the sink will
decrease under both RCP 8.5 and RCP 2.6 scenarios during the 21[st] century, while MIC-TEM
projects that the sink will increase under the RCP 8.5 but change to carbon source under the RCP
2.6 scenario. Estimations based on models without dormancy could fit observations of $R_H$ as well
as estimations with dormancy, but at the cost of underestimating microbial biomass (Wang et al.,
2014). Differences in predicted $R_H$ with and without dormancy increase with temperature and
with the length of the dry periods between wetting events (Salazar et al., 2018). The large
difference in two models suggests the importance of incorporating microbial dormancy effects.
The large bias between dormancy and non-dormancy models mainly comes from two parts.
First, many important microbial activities such as soil organic carbon decomposition and nutrient
cycling largely depend on the active fraction of microbial communities, not total microbial
biomass (Wang et al., 2014; Blagodatsky et al., 2000). However, only a small part (about 0.1-
2%, seldom exceed 5%) of the total soil microbial biomass is recognized to be active under
natural conditions (Blagodatsky et al., 2011; Werf & Verstraete, 1987). Thus, dormancy could be
a prominent feature in soil systems (Wang et al., 2014). Without considering dormancy, the
"effective" microbial biomass for soil decomposition could be overestimated, resulting in
overestimation of heterotrophic respiration (He et al., 2015). He et al. (2015) predicted total soil
$R_H$ of all temperate forests (25°N-50°N) from the dormancy model amounted to 7.28 Pg C $yr^{-1}$
and 8.83 Pg C $yr^{-1}$ from a no-dormancy model, which is 21.3% higher than the dormancy model.
Although their study region and simulation period are different from our study, the results can
still be comparable.  Both studies indicated that the magnitude of $R_H$ from no-dormancy model
are higher than dormancy models. Second, high soil respiration stimulates N mineralization in
soils (Zhuang et al., 2001, 2002), making more nutrients for photosynthesis of plants (Raich et
al., 1991; McGuire et al., 1995). Therefore, NPP will be higher due to the N enrichment from
higher $R_H$. However, how NEP will change is still unclear. Our estimates of the northern
extratropical NEP in the 1980s (1.61 Pg C $yr^{-1}$ with MIC-TEM-dormancy and 0.84 Pg C $yr^{-1}$
with MIC-TEM) are within ranges (0.6 to 2.3 PgC $yr^{-1}$) reported in the literature for northern
regions (Schimel et al.,2001).  Moreover, our predicted time trajectory  of NEP in the 21st
century under the RCP 2.6 scenario is very similar to the finding of White et al. (2000),
indicating that NEP increases from the 2000s to the 2070s, and then decreases in the 2090s.
Although our dormancy model can project reasonable carbon fluxes and indicate the importance
of incorporating microbial dormancy when compared with MIC-TEM (Zha & Zhuang et al.,
2018), there are some other microbial traits have not yet been considered in our model. For
instance, one vital common evolutionary trait of microbe is the community shift (Wang et al.,
2015) with changing environment, including warming, N fertilization and precipitation (Treseder
et al., 2011; Frey et al., 2013; Allison et al., 2009; Evans & Wallenstein, 2011). Community shift
will influence microbial physiology, temperature sensitivity and growth rates (Classen et al.,
2015), which will further affect the rate of soil decomposition and other carbon dynamics
(Treseder et al., 2011; Schimel & Schaeffer, 2012; Todd-Brown et al., 2011). Besides, microbial
community composition was ignored in our model. We didn't separate among functional
microbial groups, but gather microbes into one "box". However, microbial community
composition could influence ecosystem functioning, and their variance in responses to
environmental conditions could alter the prediction of the rates of decomposition of organic
material (Balser et al. 2002; Fierer et al. 2007). Especially, some narrowly-distributed functions
can be more sensitive to microbial community composition, and these might benefit most from
explicit consideration of distinguishing functional groups in ecosystem models (McGuire &
Treseder, 2010; Schimel 1995). Thus, functional dissimilarity in microbial communities can be
considered in next step for model development (Strickland et al., 2009; Moorhead et al., 2006).
Moreover, microbial acclimation, a mechanism of adaption to a new temperature regime, is
another important trait to affect soil decomposition. Recent studies have found that the warming-
induced elevated respiration of the microbial community could decrease over time because of
acclimation (Melillo et al. 1993; Todd-Brown et al., 2011). This mechanism shall be factored
into future soil decomposition analysis.
Except for model limitations mentioned above, additional uncertainties may come from
inadequate model parameterization and model assumptions. For example, a critical microbial
parameter, carbon use efficiency (CUE), is a primary control to soil $CO_2$ efflux. Higher CUE
indicates more microbial growth and more carbon uptake by plants, while lower CUE indicates
higher soil decomposition (Manzoni et al., 2012). Theoretical and empirical studies have
suggested that CUE depends on both temperature and substrate quality (Frey et al., 2013) and
decreases as temperature increases and nutrient availability decreases (Manzoni et al., 2012).
Our study considered the CUE sensitivity to temperature, but not nutrient availability. On the
other hand, some model assumptions can also cause uncertainties. For example, we assumed that
vegetation will not change during the transient simulation. However, over the past few decades
in northern high latitudes, temperature increases have led to vegetation shift from one type to
another (Hansen et al., 2006; White et al., 2000). The vegetation changes will affect carbon
cycling in these ecosystems.
While our analysis suggests it is important to incorporate microbial dormancy dynamics
into a process-based biogeochemistry model to more adequately simulate carbon dynamics in
northern high latitudes, we do confront modeling dilemmas.  First, our process-based models
have a relatively large number of parameters, which unavoidably creates the "equifinality"
problem as recognized in our previous studies for the model (e.g., Tang and Zhuang, 2008,
2009).  To alleviate this problem in this analysis, we have conducted parameter ensemble
simulations at both site and regional levels and presented our results with uncertainties, which
could be a standard approach for process-based complex biogeochemistry modeling analyses.
Second, incorporating more ecosystem processes increases the number of parameters in our
model, inducing even larger uncertainties for both site level and regional simulations.  On the
one hand, the more complex model to a certain degree helps capture observations, on the other
hand, the model uncertainty has not been constrained or even enlarged. We highlight the need to
further investigate this trade-off within the modeling research community.

**5. Conclusions**
This study incorporated microbial dormancy into a detailed microbial-based soil
decomposition biogeochemistry model to examine the fate of large Arctic soil carbon under
changing climate conditions.  Regional simulations using MIC-TEM-dormancy indicated that,
over the 20$^{th}$ century, the region is a carbon sink of 166.8 ± 97.7 Pg. This sink could decrease to
175.9 ± 105.4 Pg under the RCP 8.5 scenario or 125.4 ± 85.5 Pg under the RCP 2.6 scenario
during the 21$^{st}$ century. Whether considering microbial dormancy or not can cause large
differences in soil decomposition estimation between two models. Meanwhile, due to available
nitrogen affected by soil decomposition, net primary production is consequently influenced in
these two centuries. The combined changes in soil decomposition and net primary production led
to large differences in carbon budget estimation between two models. Compared with MIC-
TEM, MIC-TEM-dormancy projected 75.9 Pg more C stored in the terrestrial ecosystems over
the last century, 50.4 Pg and 125.2 Pg more C under the RCP 8.5 and RCP 2.6 scenarios,
respectively. This study highlights the importance of the representation of microbial dormancy in
earth system models in order to adequately quantify the carbon dynamics in northern high
latitudes.

**Acknowledgments**
This research was supported by a NSF project (IIS-1027955), a DOE project (DE-SC0008092),
and a NASA LCLUC project (NNX09AI26G) to Q. Z. We acknowledge the Rosen High
Performance Computing Center at Purdue for computing support. We thank the National Snow
and Ice Data center for providing Global Monthly EASE-Grid Snow Water Equivalent data,
National Oceanic and Atmospheric Administration for North American Regional Reanalysis
(NARR). We also acknowledge the World Climate Research Programme's Working Group on
Coupled Modeling Intercomparison Project CMIP5, and we thank the climate modeling groups
for producing and making available their model output. The data presented in this paper can be
accessed through our research website (http://www.eaps.purdue.edu/ebdl/)

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

**Author contributions.** Q.Z. designed the study. J.Z. conducted model development, simulation
and analysis. J.Z. and Q. Z. wrote the paper.
**Competing financial interests.** The submission has no competing financial interests.
**Materials & Correspondence.** Correspondence and material requests should be addressed to
qzhuang@purdue.edu.







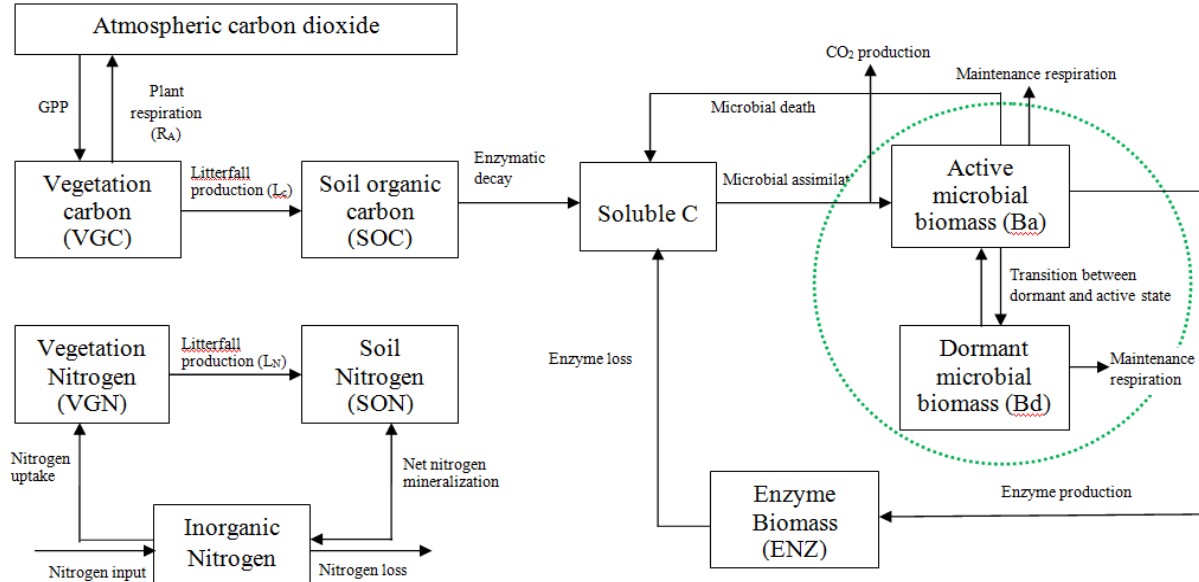

Figure 1. Framework of the dormancy model: microbial biomass is split into two parts, active
microbial biomass and dormant microbial biomass (shown in the green dashed circle).
Maintenance respiration from these two parts, and the $CO_2$ production through microbial
assimilation contributes to heterotrophic respiration. The model was revised based on Zha &
Zhuang (2018).

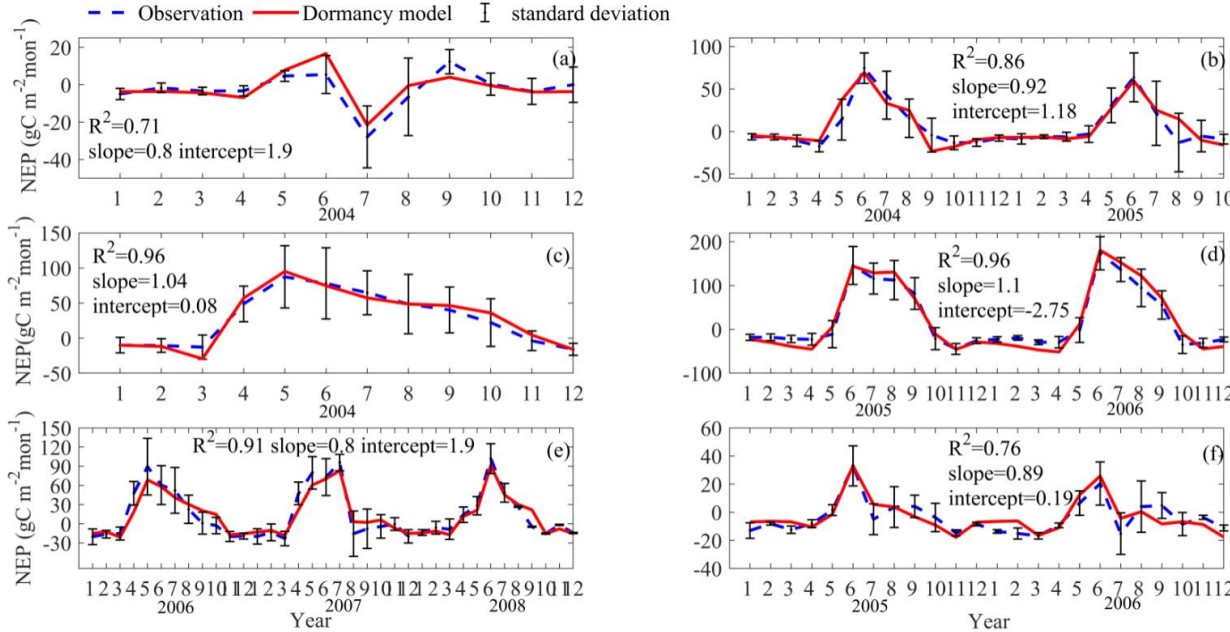

Figure 2. Comparison between observed and simulated NEP (gC m$^{-2}$mon$^{-1}$) at: (a) Ivotuk (alpine tundra), (b) UCI-1964 burn site (boreal forest), (c) Howland Forest (main tower) (temperate coniferous forest), (d) Univ. of Mich. Biological Station (Temperate deciduous forest), (e) KUOM Turfgrass Field (Grassland), and (f) Atqasuk (Wet tundra). Note: scales are different. Error bars represent standard errors among daily measure data in one month.


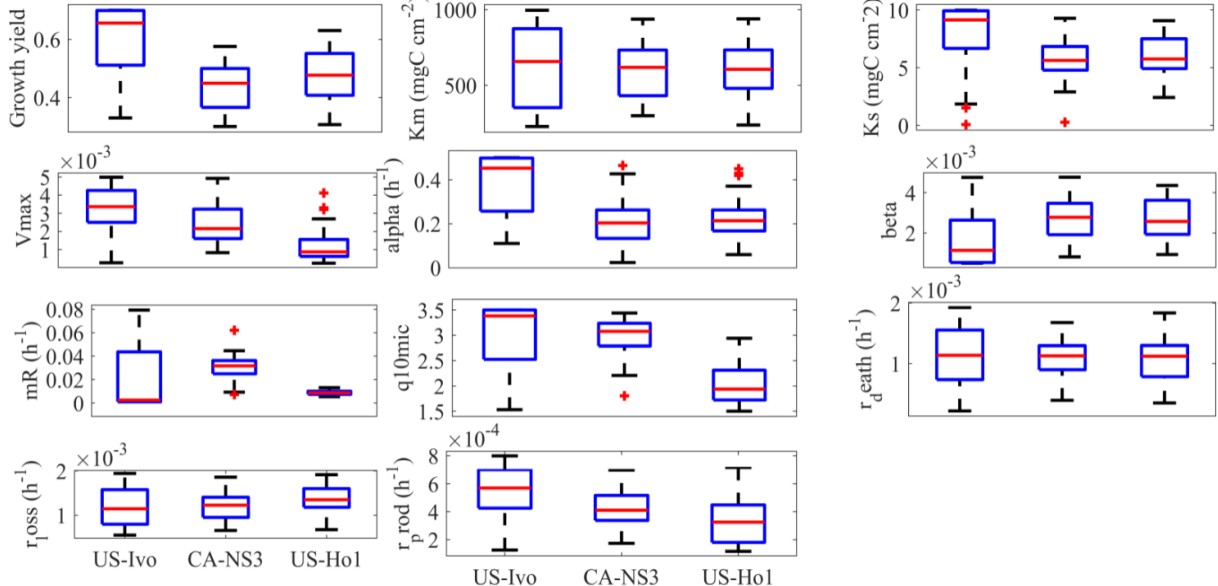

Figure 3. Boxplot of parameter posterior distribution that are obtained after ensemble inverse
modeling for MIC-TEM-dormancy all six sites: US-Ivo: Ivotuk (alpine tundra), CA-NS3: UCI-
1964 burn site (boreal forest), US-Ho1: Howland Forest (temperate coniferous forest), US-UMB:
Univ. of Mich. Biological Station (temperate deciduous forest), US-KUT: KUOM Turfgrass
Field (grassland), US-Atq: Atqasuk (wet tundra).

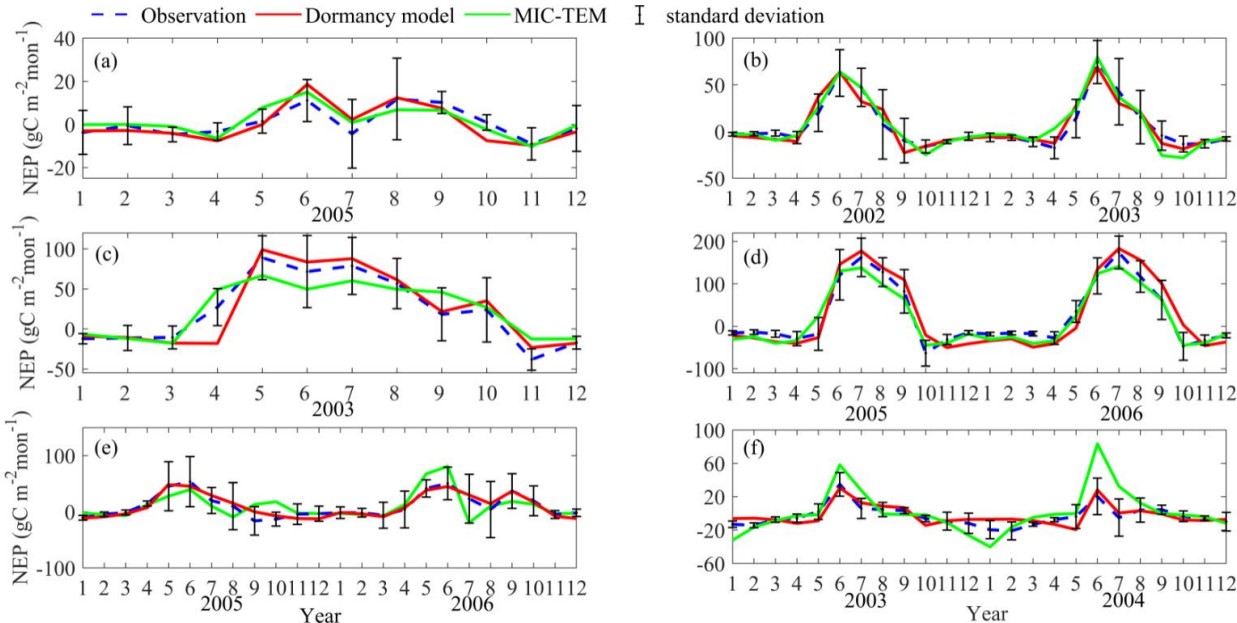

Figure 4. Comparison between observed and simulated NEP (gC m⁻²mon⁻¹) at: (a) Ivotuk (alpine
tundra), (b) UCI-1964 burn site (boreal forest), (c) Howland Forest (main tower) (temperate
coniferous forest), (d) Bartlett Experimental Forest (Temperate deciduous forest), (e) Brookings
(Grassland), and (f) Atqasuk (Wet tundra). Note: scales are different.

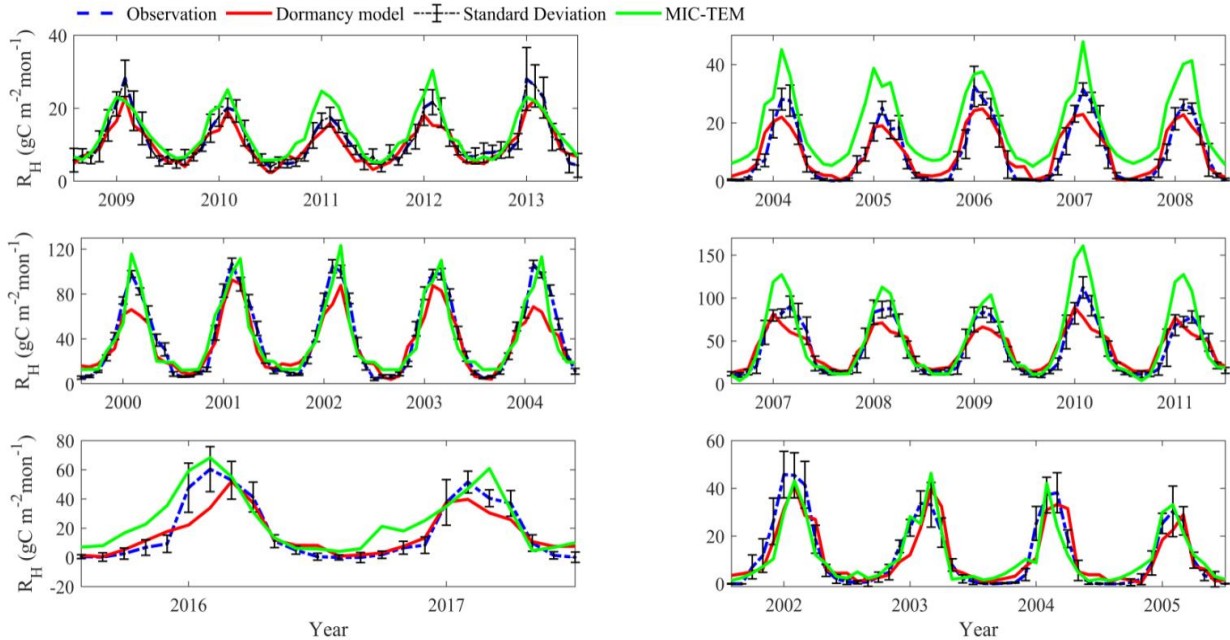

Figure 5. Comparison between observed and simulated $R_H$ (gC m$^{-2}$mon$^{-1}$) at: (a) US-EML (alpine tundra), (b) CA-SJ2 (boreal forest), (c) US-Ho2 (temperate coniferous forest), (d) US-UMB (Temperate deciduous forest), (e) US-Ro4 (Grassland), and (f) RU-Che (Wet tundra). Note: scales are different.

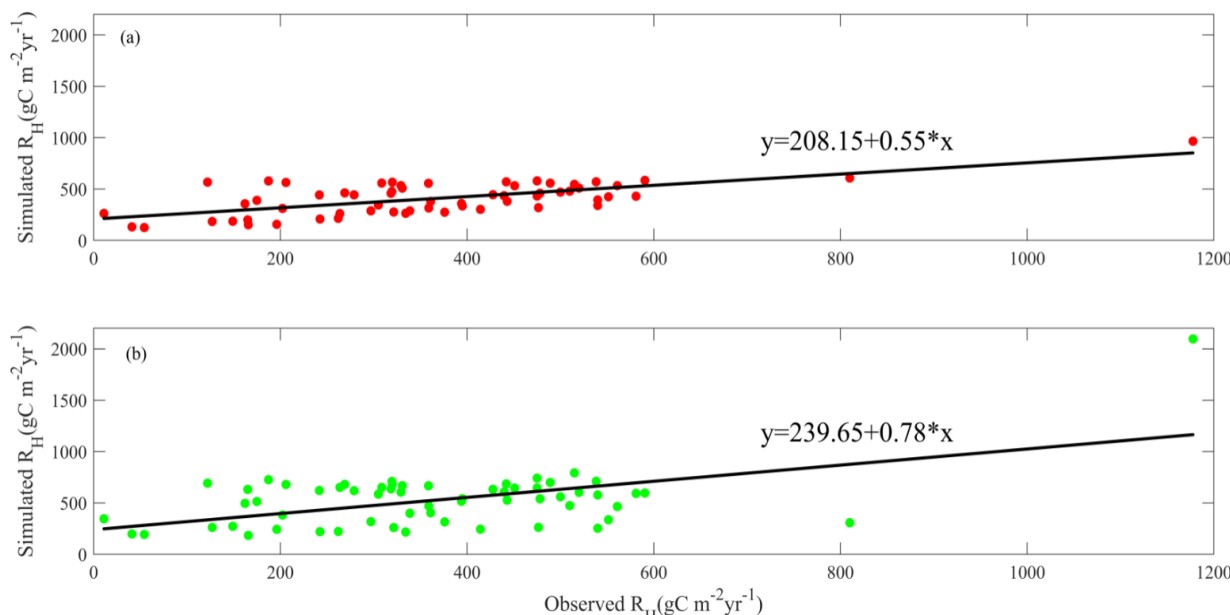

Figure 6. Linear regression between simulated and observed annual $R_H$ (gC m$^{-2}$yr$^{-1}$) for: (a) MIC-
TEM-dormancy, and (b) MIC-TEM.







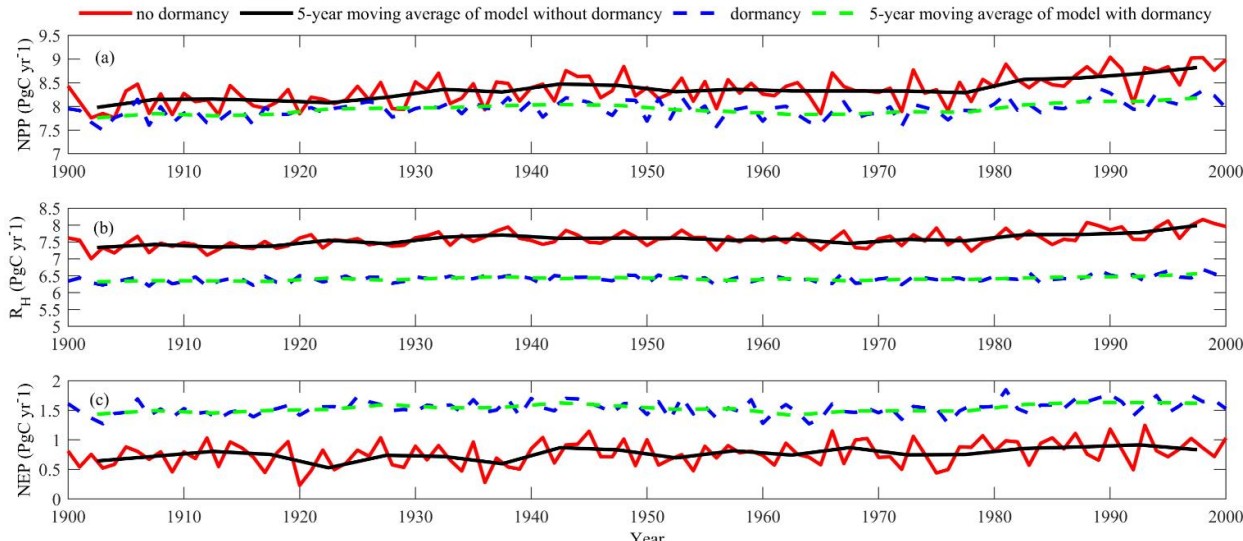


Figure 7. Simulated annual net primary production (NPP, top panel), heterotrophic respiration ($R_H$,
center panel) and net ecosystem production (NEP, bottom panel) during the 20[th] century by
dormancy model and MIC-TEM, respectively.





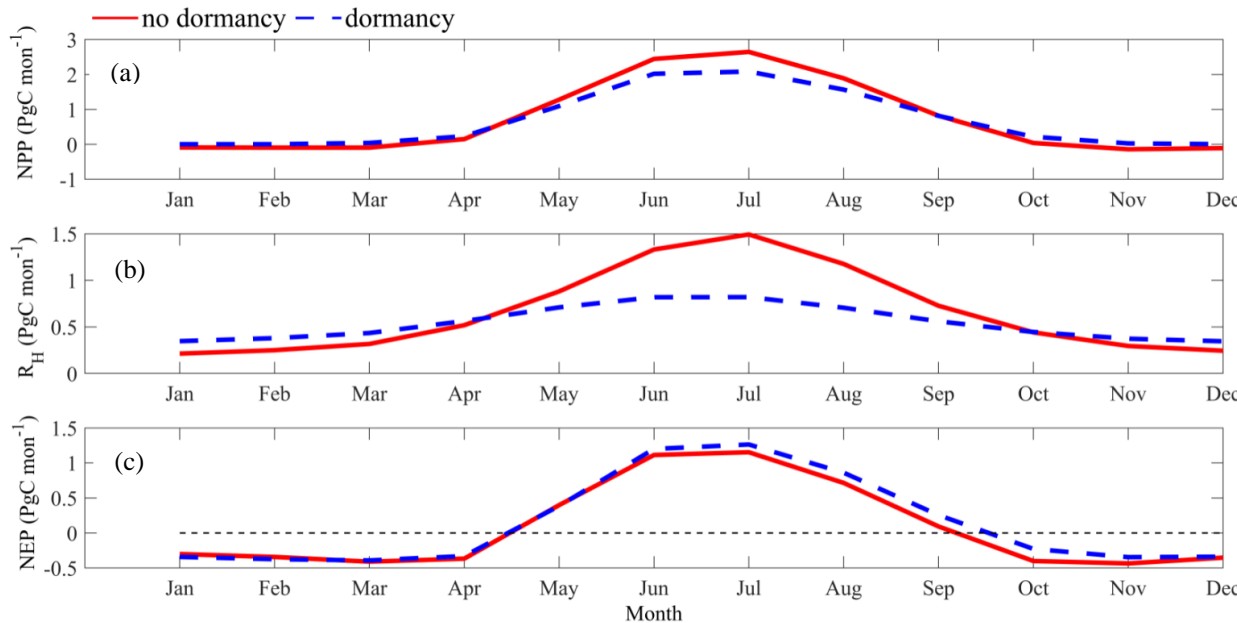


Figure 8. Regional annual seasonal pattern of simulated (a) net primary production (NPP, top
panel), (b) heterotrophic respiration ($R_H$, center panel) and (c) net ecosystem production (NEP,
bottom panel) during the 1990s from dormancy model and MIC-TEM. The region is all land
areas north of 45 ºN.










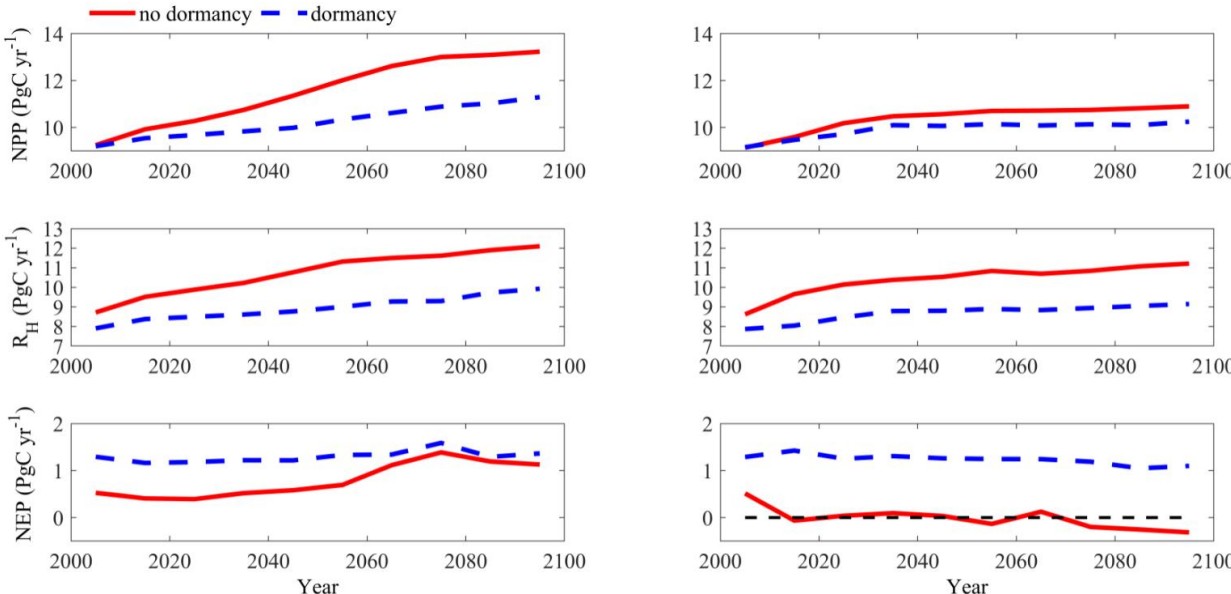


Figure 9. Predicted changes in carbon fluxes: (i) NPP, (ii) $R_H$, and (iii) NEP for all land areas north of 45 ºN in response to transient climate change under the RCP 8.5 scenario (left panel) and RCP 2.6 scenario (right panel) with dormancy model and MIC-TEM, respectively. The decadal running mean is applied.













(a)

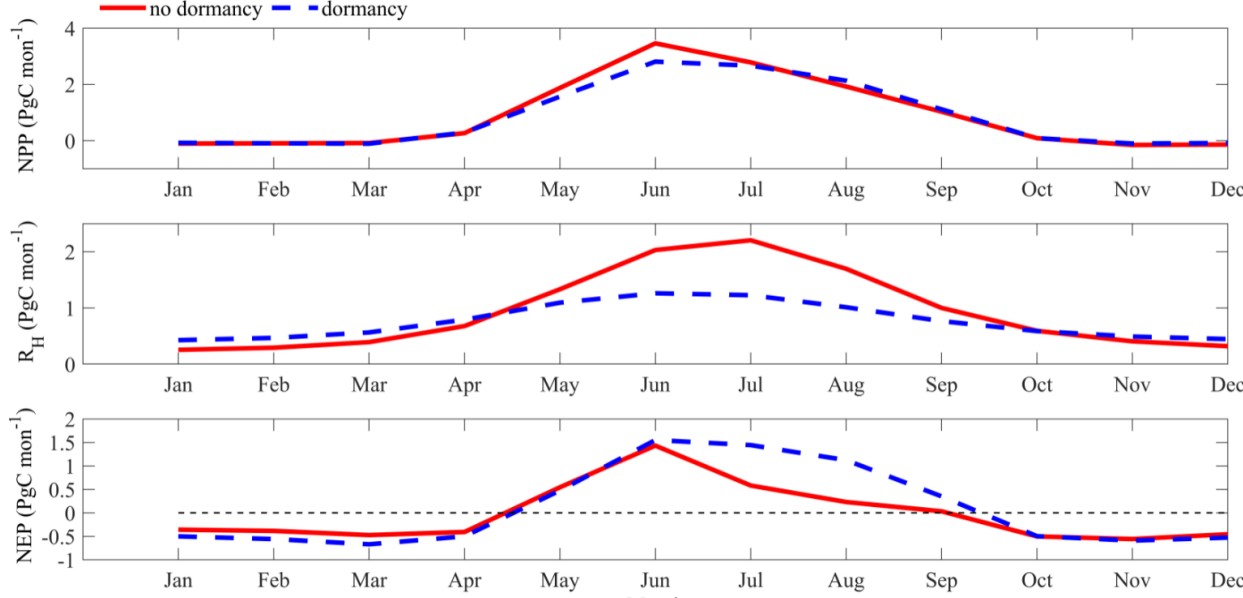


(b)

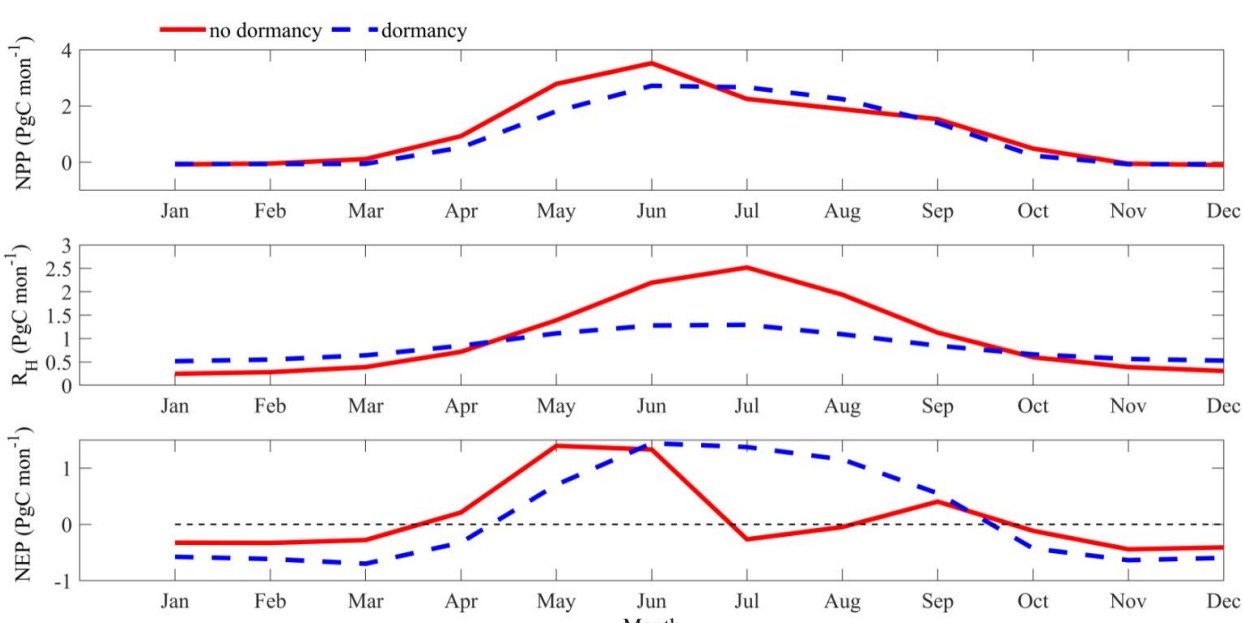


Figure 10. Regional annual seasonal pattern of simulated net primary production (NPP, top
panel), heterotrophic respiration ($R_H$, center panel) and net ecosystem production (NEP, bottom
panel) during the 2090s from dormancy model and MIC-TEM under: (a) RCP 2.6 scenario (top
panel) and (b) RCP 8.5 scenario (bottom panel). The region is all land areas north of 45 ºN.

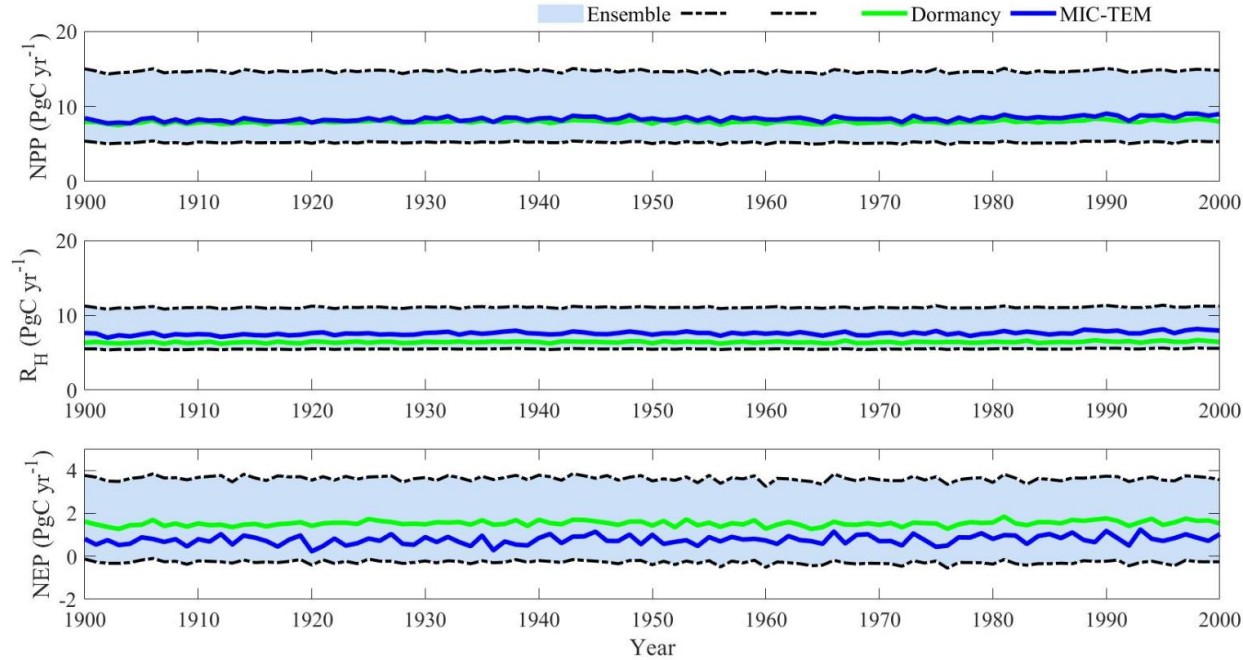

Figure 11. Simulated annual net primary production (NPP, top panel), heterotrophic respiration
(R_H, center panel) and net ecosystem production (NEP, bottom panel) by MIC-TEM-dormancy
with ensemble of parameters.

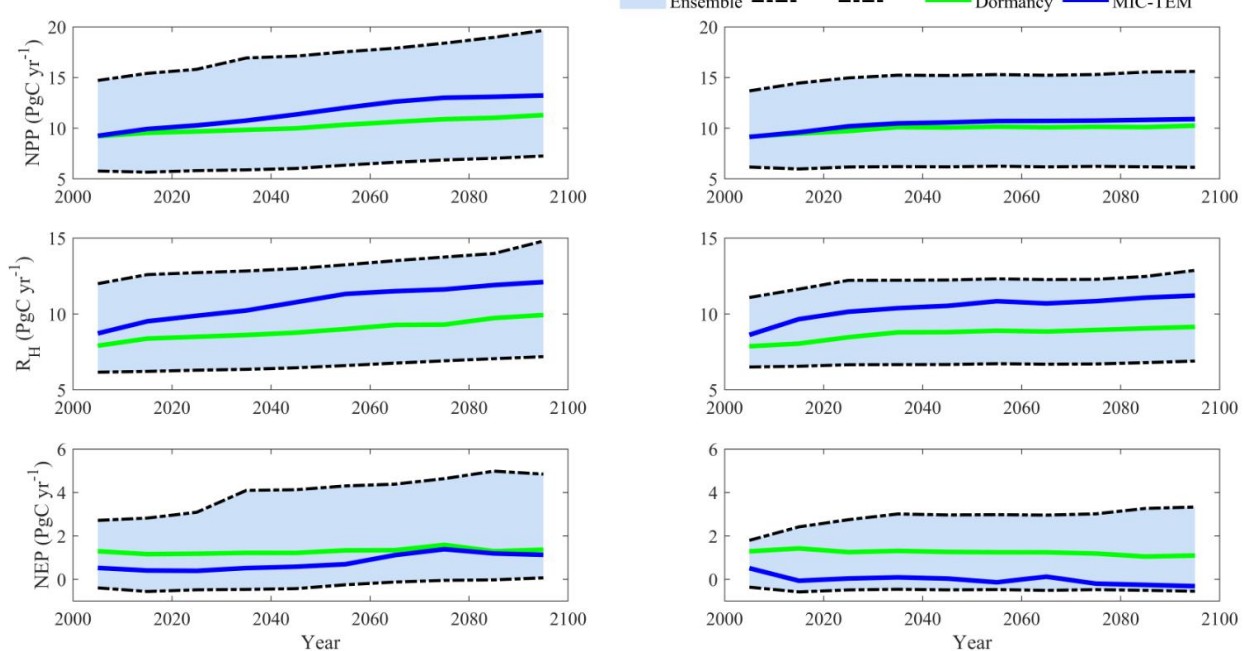

Figure 12. Simulated annual net primary production (NPP, top panel), heterotrophic respiration
(R$_H$, center panel) and net ecosystem production (NEP, bottom panel) under RCP 8.5 scenario
(left panel) and RCP 2.6 scenario (right panel) by MIC-TEM-dormancy with ensemble of
parameters. The decadal running mean is applied. The grey area represents the upper and lower
bounds of simulations.

**Table 1. Parameters associated with detailed microbial dormancy in MIC-TEM-dormancy**

| parameter | unit | description | Parameter range | references |
|---|---|---|---|---|
| $m_R$ | $h^{-1}$ | Specific maintenance rate at active state | [0.001, 0.08] | Wang et al. (2014) |
| $Q_{10mic}$ | - | Temperature effects on microbial metabolic activity (rate change per 10 °C increase in temperature). Based on 0.65 eV activation energy for soils | [1.5, 3.5] | He et al. (2015) |
| $Q_{10enz}$ | - | Temperature effects on enzyme activity (rate change per 10 °C increase in temperature). Based on 6% rate increase per degree Celsius | 1.79 | He et al. (2015) |
| $\alpha$ | - | the ratio of $m_R$ to the sum of maximum specific growth rate | [0.01, 0.5] | Wang et al. (2014) |
| $\beta$ | - | Ratio of dormant microbial maintenance rate to $m_R$ | [0.0005, 0.005] | Wang et al. (2014) |
| $Y_g$ | - | carbon use efficiency | [0.3, 0.7] | He et al. (2015) |
| $K_s$ | mgC $cm^{-2}$ | Half-saturation constant for directly accessible substrate | [0.01, 10] | Wang et al. (2014) |
| $Km_{uptake}$ | mgC $cm^{-2}$ | Half-saturation constant for enzymatic decay of SOC | [200, 1000] | He et al. (2015) |
| $r_{death}$ | $h^{-1}$ | Potential rate of microbial death | [2e$^{-4}$, 2e$^{-3}$] | Allison et al. (2010) |
| $r_{EnzProd}$ | $h^{-1}$ | Enzyme production rate of microbe | [1e$^{-4}$, 8e$^{-4}$] | He et al. (2015) |
| $r_{enzloss}$ | $h^{-1}$ | Enzyme loss rate | [0.0005, 0.002] | Allison et al. (2010) |
| $V_{max}$ | mgC $cm^{-2}$ $h^{-1}$ | Maximum SOC decay rate | [1e$^{-4}$, 5e$^{-3}$] | He et al. (2015) |

**Table 2. Site description and measured NEP data used to calibrate MIC-TEM-dormancy**

| Site Name | Location (Longitude (degrees) /Latitude (degrees)) | Elevation (m) | Vegetation type | Description | Data range | Citations |
|---|---|---|---|---|---|---|
| Univ. of Mich. Biological Station | 84.71W 45.56 N | 234 | Temperate deciduous forest | Located within a protected forest owned by the University of Michigan. Mean annual temperature is 5.83°C with mean annual precipitation of 803mm | 01/2005-12/2006 | Gough et al. (2013) |
| Howland Forest (main tower) | 68.74W 45.20N | 60 | Temperate coniferous forest | Closed coniferous forest, minimal disturbance. | 01/2004-12/2004 | Davidson et al. (2006) |
| UCI-1964 burn site | 98.38W 55.91N | 260 | Boreal forest | Located in a continental boreal forest, dominated by black spruce trees, within the BOREAS northern study area in central Manitoba, Canada. | 01/2004-10/2005 | Goulden et al. (2006) |
| KUOM Turfgrass Field | 93.19W 45.0N | 301 | Grassland | A low-maintenance lawn consisting of cool-season turfgrasses. | 01/2006-12/2008 | Hiller et al. (2011) |
| Atqasuk | 157.41W 70.47N | 15 | Wet tundra | 100 km south of Barrow, Alaska. Variety of moist-wet coastal sedge tundra, and moist-tussock tundra surfaces in the more well-drained upland. | 01/2005-12/2006 | Oechel et al. (2014); |
| Ivotuk | 155.75W 68.49N | 568 | Alpine tundra | 300 km south of Barrow and is located at the foothill of the Brooks Range and is classified as tussock sedge, dwarf-shrub, moss tundra. | 01/2004-12/2004 | McEwing et al. (2015) |


**Table 3. Site description and measured NEP data used to validate MIC-TEM-dormancy**

| Site Name | Location (Longitude (degrees) /Latitude (degrees)) | Elevation (m) | Vegetation type | Description | Data range | Citations |
|---|---|---|---|---|---|---|
| Bartlett Experimental Forest | 71.29W/ 44.06N | 272 | Temperate deciduous forest | Located within the White Mountains National Forest in north-central New Hampshire, USA, with mean annual temperature of 5.61 °C and mean annual precipitation of 1246mm. | 01/2005-12/2006 | Jenkins et al. (2007); Richardson et al. (2007); |
| Howland Forest (main tower) | 68.74W/ 45.20N | 60 | Temperate coniferous forest | Closed coniferous forest, minimal disturbance. | 01/2003-12/2003 | Davidson et al. (2006) |
| UCI-1964 burn site | 98.38W/ 55.91N | 260 | Boreal forest | Located in a continental boreal forest, dominated by black spruce trees, within the BOREAS northern study area in central Manitoba, Canada. | 01/2002-12/2003 | Goulden et al. (2006) |
| Brookings | 96.84W/ 44.35N | 510 | Grassland | Located in a private pasture, belonging to the Northern Great Plains Rangelands, the grassland is representative of many in the north central United States, with seasonal winter conditions and a wet growing season. | 01/2005-12/2006 | Gilmanov et al. (2005) |
| Atqasuk | 157.41W/ 70.47N | 15 | Wet tundra | 100 km south of Barrow, Alaska. Variety of moist-wet coastal sedge tundra, and moist-tussock tundra surfaces in the more well-drained upland. | 01/2003-12/2004 | Oechel et al. (2014); |
| Ivotuk | 155.75W/ 68.49N | 568 | Alpine tundra | 300 km south of Barrow and is located at the foothill of the Brooks Range and is classified as tussock sedge, dwarf-shrub, moss tundra. | 01/2005-12/2005 | McEwing et al. (2015) |

**Table 4. Site description and measured $R_H$ data used to validate MIC-TEM-dormancy model**

| Site | Location (Longitude (degrees) /Latitude (degrees)) | Elevation (m) | Vegetation type | Data range | Citations |
|---|---|---|---|---|---|
| US-EML | 149.25W/ 63.88N | 700 | Alpine tundra | 01/2009- 12/2013 | Belshe et al. (2012) |
| CA-SJ2 | 104.65W/ 53.95N | 580 | Boreal forest | 01/2004- 12/2008 | Coursolle et al. (2006) |
| US-Ho2 | 68.75W/ 45.21N | 91 | Temperate coniferous forest | 01/2000- 12/2004 | Davidson et al. (2006) |
| US-UMB | 84.71W/ 45.56N | 234 | Temperate deciduous forest | 01/2005- 12/2006 | Gough et al. (2013) |
| US-Ro4 | 93.07W/ 44.68N | 274 | Grasslands | 01/2016- 12/2017 | Griffis et al. (2011) |
| RU-Che | 161.34E/ 68.61N | 6 | Wet tundra | 01/2002- 12/2005 | Merbold et al. (2009) |

**Table 5. Model validation statistics for Dormancy model and MIC-TEM at six sites with NEP data**

| Site Name | Vegetation type | Models | Intercept | Slope | R-square | Adjusted R-square | p-value |
|---|---|---|---|---|---|---|---|
| Ivotuk | Alpine tundra | MIC-TEM | 0.85 | 0.83 | 0.70 | 0.67 | <0.001 |
| | | Dormancy | -0.51 | 1.09 | 0.75 | 0.73 | <0.001 |
| UCI-1964 burn site | Boreal forest | MIC-TEM | 0.18 | 1.03 | 0.912 | 0.9080 | <0.001 |
| | | Dormancy | -0.21 | 0.96 | 0.90 | 0.894 | <0.001 |
| Howland Forest (main tower) | Temperate coniferous forest | MIC-TEM | 7.29 | 0.72 | 0.85 | 0.83 | <0.001 |
| | | Dormancy | 0.27 | 1.05 | 0.89 | 0.88 | <0.001 |
| Bartlett Experimental Forest | Temperate deciduous forest | MIC-TEM | -6.05 | 0.91 | 0.944 | 0.941 | <0.001 |
| | | Dormancy | -2.34 | 1.13 | 0.93 | 0.924 | <0.001 |
| Brookings | Grassland | MIC-TEM | 3.05 | 0.71 | 0.84 | 0.83 | <0.001 |
| | | Dormancy | 0.17 | 0.95 | 0.90 | 0.898 | <0.001 |
| Atqasuk | Wet tundra | MIC-TEM | 7.22 | 1.85 | 0.71 | 0.70 | <0.001 |
| | | Dormancy | 0.19 | 0.82 | 0.67 | 0.66 | <0.001 |

**Table 6. Model validation statistics for Dormancy model and MIC-TEM at six sites with $R_H$ data**

1063

| Site ID | Vegetation type | Models | Intercept | Slope | R-square | Adjusted R-square | RMSE | p-value |
|---------|-----------------|--------|-----------|-------|----------|-------------------|------|---------|
| US-EML | Alpine tundra | MIC-TEM | 2.90 | 0.91 | 0.79 | 0.78 | 3.55 | <0.001 |
|  |  | Dormancy | 1.81 | 0.74 | 0.87 | 0.85 | 2.69 | <0.001 |
|  |  |  |  |  |  |  |  |  |
| CA-SJ2 | Boreal forest | MIC-TEM | 7.59 | 1.12 | 0.84 | 0.83 | 9.8 | <0.001 |
|  |  | Dormancy | 2.6 | 0.74 | 0.86 | 0.85 | 3.97 | <0.001 |
|  |  |  |  |  |  |  |  |  |
| US-Ho2 | Temperate coniferous forest | MIC-TEM | 4.07 | 0.89 | 0.86 | 0.84 | 12.39 | <0.001 |
|  |  | Dormancy | 6.59 | 0.71 | 0.91 | 0.89 | 11.83 | <0.001 |
|  |  |  |  |  |  |  |  |  |
| US-UMB | Temperate deciduous forest | MIC-TEM | -4.73 | 1.32 | 0.81 | 0.8 | 20.05 | <0.001 |
|  |  | Dormancy | 13.6 | 0.67 | 0.85 | 0.84 | 12.94 | <0.001 |
|  |  |  |  |  |  |  |  |  |
| US-Ro4 | Grassland | MIC-TEM | 9.34 | 0.87 | 0.81 | 0.79 | 11.25 | <0.001 |
|  |  | Dormancy | 4.81 | 0.65 | 0.86 | 0.84 | 9.21 | <0.001 |
|  |  |  |  |  |  |  |  |  |
| RU-Che | Wet tundra | MIC-TEM | 2.5 | 0.67 | 0.72 | 0.71 | 6.24 | <0.001 |
|  |  | Dormancy | 1.96 | 0.77 | 0.81 | 0.79 | 5.95 | <0.001 |