# Peer review of "Microbial dormancy and its impacts on Arctic terrestrial ecosystem carbon budget Junrong Zha and Qianlai Zhuang Department of Earth, Atmospheric, and Planetary Sciences and Department of Agronomy, Purdue University, West Lafayette, IN 47907 USA Submitted to: Biogeoscience Correspondence to: qzhuang@"

_Biogeosciences, 2019_

## Referee Comment (RC1) · Thomas Wutzler (Referee) · 23 May 2019

Although, I have some concerns with the model formulation, I will focus on the calibration and validation. For the model formulation some more background and motivation of the used formulation would be appreciated. Currently, I need to read the referenced He paper to understand it.

I was happy to read about the model validation at independent sites and data. However, the calibration/validation it is not yet presented in a way to fully understand it and uncertainties are lacking. Uncertainties are necessary to answer my most important concern: Is the introduction of several more degrees of freedom, i.e. free parameters, justified? Comparing predictions based on a single optimal parameter set is not

sufficient to justify a more complex model.

Fig. 3 shows that many parameters are poorly constrained (non-experts should be provided to compare the prior in Fig. 3 to the posterior see this). The problem of more degrees of freedom with more details models is equifinality: several combination of parameters match the data similarly well. That needs to be incorporated in forward simulations, which usually become more uncertain with equifinality.

The simulations need to be repeated for both model versions with a larger number of viable parameters sets - and in the validation need to include also the variation of parameters that were not part of the current calibration. Then the 95% confidence bounds can be displayed in the comparison to the data - at least for a few sites. I suspect that adding too much detail will increase the confidence bounds until 2100 so wide, that the dormancy model (and maybe even the TEM-MIC) cannot generate conclusive trajectories. But the authors may prove me wrong.

Detailed comments

Fig 3: It is not clearly stated, how many parameters were calibrated and Fig. 3 is barely readable because of display quality. Are there only 3 out of the 6 sites displayed? For the moment, I assume that the parameters of Table 1 were calibrated. I am also confused that you show distributions separately by site. As I understand, you need a cross-site parameter set. Which parameters (also of the non-optimized ones) differ by stratum (site/pixel/vegetation type/soil type)?

Cost function (17): Why did you not consider uncertainty of observed NEE? Usually, you need this to determine, which parameter sets are viable. If you have larger NEE confidence bounds, also more different parameter sets will generate predictions that are still compatible with the calibration NEE. For my main concern above it is important to keep also the slightly less optimal but compatible parameter sets.

L 294: Taking the mean of parameter sets is not recommended. If the parameter sets

belong to the same limiting distribution, I'd rather pick one random sample if I forward runs with with a larger set are too expensive. For a proper analysis you need to run forward the model with several different parameter sets and then compute the mean and confidence bounds of the predictions. If parameters converged to a few several limiting distributions, i.e. clusters in the parameter sets, you need to report also follow ups for these alternative optima. If parameters did not converge, you have not yet successfully optimized the model.

The conclusion that a model of more degrees of freedom and re-calibration fits the observations better is plain to me, and not enough for a justification for using this model.

---

## Referee Comment (RC2) · Alejandro Salazar (Referee) · 12 Sep 2019

General comments

Although no questions, hypotheses or goals are explicitly described in this manuscript, the authors implicitly addressed interesting and novel questions e.g. what is the role of microbial dormancy in the carbon budget of the Arctic terrestrial ecosystems? How do projections of net primary production (NPP), soil heterotrophic respiration (RH) and net ecosystem production (NEP) change when considering microbial dormancy in soil bio-geochemistry models? For the most part (see specific comments about mathematical formulations), the methods to address these implicit questions are valid and well explained. The authors found that the fit between observations of NPP, RH, and NEP and

model predictions with the soil biogeochemistry model MIC-TEM, is better when microbial dormancy is considered (i.e. MIC-TEM-dormancy). Moreover, predictions from MIC-TEM and MIC-TEM-dormancy varied notably across seasons and under RCP 2.6 and 8.5 scenarios, suggesting that microbial dormancy plays and important role in the ways soils from arctic terrestrial ecosystems respond to seasonal and global warming. For example, during winter MIC-TEM does not account for maintenance respiration of microbes in dormant state, while MIC-TEM-dormancy does. This leads to predictions of larger winter RH by MIC-TEM-dormancy than by MIC-TEM. In contrast, during summer MIC-TEM assumes that all soil microbes are metabolically active while MIC-TEM-dormancy acknowledges that (as empirical evidence show) in natural soils a large proportion of microbes remain in dormant metabolic state, even in summer. This because of limitations in factors other than temperature, such as moisture and nutrients. Therefore, MIC-TEM predicts (likely unrealistically) larger summer RH than MIC-TEM-dormancy. Although overall good, this work suffers from two main weaknesses: 1. The authors almost completely ignored in their discussion previous efforts to include microbial dormancy in soil biogeochemistry models. It would be interesting, for example, to know what is the contribution of this work to those previous efforts; and what are the consistencies/inconsistencies of their findings in comparison with results from other "dormancy models". 2. The language. There are so many small language issues that make the reading of this manuscript hard. The authors jump between active and passive voice, and between past and present tense. They explain abbreviations over and over again (RH was defined four times!). They show some data in the results and then show it again in the discussion. I added many small comments that in my opinion could help to address these and other similar issues.

Specific comments

L93: consider replacing "to remedy the inadequate representation of soil decomposition process" with something like: to represent decomposition in ways that include important microbial processes that were previously ignored

L100: ...models (Wieder et al., 2015)

L107: "a fraction of number of microbes, likely below 50% of live microbes, in natural soils" do you mean: a fraction, likely below 50%, of metabolically active microbes in natural soils ?

L112: ...biomass (Wieder et al., 2015)

L111: use total rather than active microbial biomass as indicator of microbial activities, which could...

L114: modeling microbial dormancy in the Arctic is important not only because arctic ecosystems are N-limited (many other places in the world are N-limited as well (Wang et al., 2010. BG), but also (and maybe more importantly) because of the marked seasonality (i.e. activity/dormancy cycles) and the above-global-average warming happening in those latitudes (which could increase the abundance/proportion of active microbes in soil). You could add this points here.

L118: could lead to better projections of... Also, what do you mean by "better"? e.g. increasing realism in a model does not necessarily increase certainty (Sulman et al., 2018. Biogeochemistry)

L124: No explicit question, hypotheses and/or goals? It would be very interesting and useful to know what where the authors expecting, and why, before collecting their data. If there were no hypotheses driving this study, it would be too late now to formulate them.

L129-124: This paragraph seems like a good place to mention previous attempts to include microbial dormancy into process-based biogeochemistry model - other than in MIC-TEM (He at al., 2015) – e.g. in MEND (Wang et al., 2015. ISME) and CORPSE (Salazar et al., 2018. SBB)

L130-133: "First, we describe..." (first person, present tense) "Second, parameterization and validation of... have been shown" (third person) "Third, we applied..." (first

person, past tense) Unify

L138: I don't understand the use of citation "(Zhuang et al., 2001, 2002, 2003)" here.

L148: in the new model (Figure 1), which was ignored in MIC-TEM

L155: In MIC-TEM-dormancy… We already know that this is the new model (L122 and 128)

L170: represents microbial assimilation…

L173: is maintenance weight

L173-174: "Here.. quality" odd. Maybe …biomass. CNsoil and CNmic account for substrate quality

L174: $\Phi$ is the substrate saturation level

L183: Why is the equation for Dliq not numbered?

L186: Replace "We used… denotes" for "Where… denote", and move sentence to L191, after equations 7 and 8

L191: "Dormancy rate is affected by substrate availability (Ba, Bd)" this is very confusing. Ba and Bd are active and dormant biomass! (L193 and 202)

L203: Dormant microbes are tough but, no death?

L223: "is different" vague.

L226: RH already defined in L150.

L227: Consider replacing "will also be affected" by: can too, or something like that.

L249: Again, RH already defined in L150.

L250: Because of limitations in the amount of available RH data

L256: See comment in L155

L279: "were also used. In our model, we assumed" Constant jumps between active and passive voice. Choose one and stick to it. I'd recommend active voice.

L295: CUE was already defined in L152. Also, what do you mean with "was much higher in tundra types than in forests" in Figure 3? None of the boxplots in Figure 3 has CUE on the y axis.

L298: What do you mean by "much higher"? Did you test if those differences are statistically significant? At first glance, it doesn't seem so.

L299: Similar than the previous comment, what do you mean with "The opposite can be seen from parameter beta"? Although there are no names in the x-axis in the boxplot for beta in Figure 3 (see comment in L841), at first glance there seem to be no (statistically significant) difference in beta between these three sites. Again, did you test this?

L305: Which statistical analysis? There is no mention of any statistical analysis in the methods.

L308: Delete "which is no dormancy-based". We have this clear by now!

L310: "Another set of sites..." So, data from Figure 6 is not the same than from Figure 5? Why using "other set of sites"? Which sites?

L311: "The comparisons between monthly observed RH and simulated RH from two contrasting models were conductd" delete. Redundant. If you decide to keep it, fix typo: "conductd".

L313: root mean square error (RMSE)

L316: RH already defined in L150.

L317: "This difference further affects soil available nitrogen dynamics, influencing nitrogen uptake by plants, the rate of photosynthesis and NPP." How? Any result or reference to support this claim?

L322: "Here positive values of NEP represent sinks of CO2 into terrestrial ecosystems, while negative values represent sources of CO2 to the atmosphere." Already mentioned in L286-288.

L327: Consider deleting "which estimates 75.9 Pg more carbon sink than MIC-TEM does but with less interannual variation (Figure 7c)". 1. Values for this and previous two sentences are not shown in Figure 7c; 2. Makes the entire sentence long and confusing.

L331: "except a slight decrease during the 1960s (Figure 7)" I don't see this.

L332: "MIC-TEM-dormancy estimated NPP and RH at 7.94 Pg C yr-1 and 6.4 Pg C yr-1 333 , which are 5.8% and 16.3% less than the estimations from MIC-TEM, respectively (Figures 7a and 7b)" Interesting!

L334: "This pronounced difference of NEP between two models comes from the disparity between the simulated NPP and RH with them." Confusing

L338: "work for soil decomposition" odd. Maybe: can decompose organic matter.

L351: Consider deleting this "This is because higher RH can cause higher NPP due to the reasons we have mentioned above."

L353: "since it's the difference between NPP and RH" either delete or move to introduction or methods (depending on how you frame it).

L355-358: Redundant with L343-349. Combine/synthesize them.

L361: "predicted that the sink is 129.9 Pg" change for something like: predicts a net C sequestration of 129.9 Pg by the end of this century. Same comment for "estimates the sink is 79.5". Also, note the inconsistency in the use of tenses: "predicted" (past), "estimates" (present)

L365: Change "MIC-TEM but with" for "MIC-TEM, with".

[Figure]

L365: Consider adding: under this scenario, "both models..."

L370: Start new paragraph with "Under the RCP 2.6 scenario..."

L375: an interannual

L381: "Similar seasonal cycle pattern appears for NPP projection". Not quite so. NPP is the same in winter with or without dormancy, and in the late summer is higher with than without dormancy (i.e. opposite to RH), especially in the RCP 8.5 scenario.

L384: "but similar NEP in other months to MIC-TEM (Figure 10)." Estimation of NEP from January to April seem lower with than without dormancy.

L388: Seems more like the opening of an introduction than of a discussion.

L390: is currently stored? Also, delete "region" or rephrase (e.g. latitude regions).

L391: "climate over this region has warmed in recent decades" other regions are warming too! Maybe you want to say that the magnitude of the warming in these regions is larger, almost twice, that of the global average.

L398: This seems like a god place to discuss your results in comparison with "results from other process-based models". e.g. - Estimations based on models without dormancy could fit observations of RH as well as estimations with dormancy, but at the cost of underestimating microbial biomass (i.e. the right result for the wrong reasons) (Wang et al., 2014. ISME) - Effects of warming on RH (and microbial biomass) may depend on factors not explicitly considered in your study e.g. differences in predicted RH with and without dormancy increase with temperature AND with the length of the dry periods between wetting events (Salazar et al., 2018. SBB).

L406: What do you mean by "most important microbial activities".

L413-419: Delete. Repeated information presented in results section.

L423: What do you mean by "and proportion"?

L427. Delete "Our regional estimate of NEP during the 20th century by MIC-TEM-dormancy is 1.54 Pg C yr-1, and is 0.78 Pg C yr-1 by MIC-TEM." Repeated information presented in results section.

L428-432: Replace "Schimel et al. (2001) reported that a range of estimates of the northern extratropical NEP is from 0.6 to 2.3 PgC yr-1 in the 1980s. In comparison with our estimates of 1.61 Pg C yr-1 430 with MIC-TEM-dormancy and 0.84 Pg C yr-1 with MIC-TEM, our regional estimates of NEP are in reasonable range." with something like: our estimates of... are within ranges reported in the literature for northern regions (estimates; Schimel).

L432: "our predicted trend of NEP is very similar to the finding of White et al. (2000), indicating that NEP increases from the 2000s to the 2070s, and then decreases in the 2090s" Which trend? "Trends" in your simulations are very different between models and between RCP scenarios!

L434: "future simulations"? Delete "future"

L434: Delete "future simulations under two contrasting climate scenarios (RCP 2.6 and RCP 8.5) exhibit a large difference of 81.1 Pg C of cumulative NEP during the 21st century by MIC-TEM, but only 6.3 Pg C of that by MIC-TEM-dormancy." Again, these are results, not a discussion. Also, "future simulations"?

L439: "no dormancy model (MIC-TEM" At this point we know this very well! No need to explain again that MIC-TEM is the "no dormancy model"

L448: "Recent studies have found the capacity of the microbial community to maintain the warming-induced elevated respiration could decrease over time because of acclimation" I don't understand this sentence.

L451: You talk about "community composition" (L442-446), move on to acclimation (L447-451), and then go back to "community composition" (L451-460). Rearrange this paragraph to avoid this jumping back.

L461: "above model limitations"? Do you mean: model limitations mentioned above?

L473: Include somewhere in the discussion: How does your work contribute to previous efforts to include microbial dormancy in soil biogeochemistry models (others than MIC-TEM; e.g. MEND, Wang et al., 2015. ISME; CORPSE, Salazar et al., 2018. SBB)? Consistencies/inconsistencies between your "dormancy model" versus other "dormancy models"?

L848: Consider deleting Table 5 and figure 6 and show summary of stats here, as you do in Figure 2

L854: Consider deleting Table 5 and figure 6 and show summary of stats here, as you do in Figure 2

L859: If you decide to keep this figure, consider deleting Table 5 and show stat data (e.g. R2) here.

Technical corrections

L123: In some places the font seems smaller than in the rest of the ms e.g. in L123: "in the Arctic terrestrial ecosystems (north 45 °N above)".

L173: ratios

L309: "Both" instead of "two"

L314: regions

L339: overestimation

L343: "both two" delete "two"

L577: Wang, G… (and move down in the ref. list)

L841: x-axis in right boxes

L863: Close parentheses in y axes

---

## Author Comment (AC1) · 15 Jan 2020

Junrong Zha and Qianlai Zhuang

jzha@purdue.edu

We thank the Associate Editor and two referees for their providing constructive comments to this manuscript. Below we detail how we have revised the manuscript following their suggestions. 1. The problem of more degrees of freedom with more details models is equifinality: several combinations of parameters match the data similarly well. That needs to be incorporated in forward simulations, which usually become more uncertain with equifinality. Response: Thanks for the comments. To address the impacts of "equifinality" on our quantifications associated with parameters, we have conducted ensemble simulations for both 20th and 21st centuries with respect to uncertain parameters. These ensemble simulations shall cover the "equifinality" set of parameters in our model. In other words, these simulations shall have included the

"equifinality" impacts. We presented the simulations results in Figure 11 and Figure 12 in this revision. 2. Fig 3: It is not clearly stated, how many parameters were calibrated and Fig. 3 is barely readable because of display quality. Are there only 3 out of the 6 sites displayed? Response: Thanks for the comments. We have revised Figure 3. Now six sites are shown andthe figure shall be more readable. 3. Cost function (17): Why did you not consider uncertainty of observed NEE? Usually, you need this to determine, which parameter sets are viable. If you have larger NEE confidence bounds, also more different parameter sets will generate predictions that are still compatible with the calibration NEE. For my main concern above it is important to keep also the slightly less optimal but compatible parameter sets. Response: The error or uncertainty of the NEE data we used have not been provided by field experimentalists. Thus, in this study, the model parameters are only constrained by the observed magnitudes and temporal variabilities of NEE at those sites.

[Figure]

**Fig. 1.**

[Figure]

**Fig. 2.**

---

## Author Comment (AC2) · 15 Jan 2020

Junrong Zha and Qianlai Zhuang

jzha@purdue.edu

We thank the Associate Editor and two referees for their providing constructive comments to this manuscript. Below we detail how we have revised the manuscript following their suggestions. 1. The authors almost completely ignored in their discussion previous efforts to include microbial dormancy in soil biogeochemistry models. It would be interesting, for example, to know what is the contribution of this work to those previous efforts; and what are the consistencies/inconsistencies of their findings in comparison with results from other "dormancy models". Response: Thanks for the comments. We have strengthened the discussion by comparing our results with previous ones in the revised manuscript. 2. The language. There are so many small language issues that make the reading of this manuscript hard. The authors jump between active and pas-

sive voice, and between past and present tense. They explain abbreviations over and over again (RH was defined four times!). They show some data in the results and then show it again in the discussion. I added many small comments that in my opinion could help to address these and other similar issues. Response: Thanks for the comments. We have carefully revised the language according to your comments one by one.

———————————————

---

## Author Comment (AC3) · 15 Jan 2020

We thank the Associate Editor and two referees for their providing constructive comments to this manuscript. Below we detail how we have revised the manuscript following their suggestions.

1. The problem of more degrees of freedom with more details models is equifinality: several combinations of parameters match the data similarly well. That needs to be incorporated in forward simulations, which usually become more uncertain with equifinality.

*Response: Thanks for the comments. To address the impacts of "equifinality" on our*

*quantifications associated with parameters, we have conducted ensemble simulations for both*

*$20^{th}$ and $21^{st}$ centuries with respect to uncertain parameters.  These ensemble simulations shall*

*cover the "equifinality" set of parameters in our model.  In other words, these simulations shall*

*have included the "equifinality" impacts. We presented the simulations results in Figure 11 and*

*Figure 12 in this revision.*

2. Fig 3: It is not clearly stated, how many parameters were calibrated and Fig. 3 is barely readable because of display quality. Are there only 3 out of the 6 sites displayed?

*Response: Thanks for the comments. We have revised Figure 3. Now six sites are shown andthe*

*figure shall be more readable.*

3. Cost function (17): Why did you not consider uncertainty of observed NEE? Usually, you need this to determine, which parameter sets are viable. If you have larger NEE confidence bounds, also more different parameter sets will generate predictions that are still compatible with the calibration NEE. For my main concern above it is important to keep also the slightly less optimal but compatible parameter sets.

*Response: The error or uncertainty of the NEE data we used have not been provided by field*

*experimentalists.  Thus, in this study, the model parameters are only constrained by the observed*

*magnitudes and temporal variabilities of NEE at those sites.*

[revised manuscript text omitted]

under simulated cycles of stressful (dryness) and favorable (wet pulses) conditions.  Our study extend those modeling studies to the whole Arctic region by developing a more detailed biogeochemistry model considering the dormancy impacts. In Below, this paper, fFirst, we first describe how we developedd the new model (MIC-TEM-dormancy) by incorporating the microbial dormancy trait into an existing microbial-based biogeochemistry model (MIC-TEM).

Second, we discuss how conduct the parameterization and validation of MIC-TEM-dormancy model were conducted using observed net ecosystem exchange data, and heterotrophic respiration data at representative sites have been shown. Third, we presented how applyied the model was applied to to northern high latitudes (above 45 ºoN) for the 20$^{th}$ and 21$^{st}$ centuries and , discussed to demonstrate 
[revised manuscript text omitted]

---

## Editor Comment (EC1) · Jens-Arne Subke (Editor) · 24 Jan 2020

Dear Drs Zha and Zhuang,

Thank you for responding to the referee reports. Work on the manuscript has addressed many of the concerns by referees, and I am content with these (e.g. the issue of equifinality). However, I am not convinced that you have managed to alleviate the main point of referee 1, which is that the uncertainty associated with both models compared here is so large that it is not possible to judge any potential improvement from incorporating dormancy. I would therefor like to ask you to revise the manuscript further to address the issue of modelling uncertainty. I include some further specific explanations below.

I agree with referee 1 that model uncertainty is key for this comparison, and this is not well illustrated also in the revised paper. Figures 2, 4 and 5 show that in some instances the new model follows temporal dynamics better, but the improvement over the non-dormancy model is not that clear, and in absence of explicit uncertainty associated with either model, the reader can not judge whether this is a significant improvement. This then leads to similar problems when making regional predictions – Are stated differences for the contrasting models within the model uncertainty?

You also don't address fully the point that the dormancy model increases modelling uncertainty owing to the larger number of parameters. As no modelling uncertainty is quantified at present, I think that this fundamental limitation of constructing a more complex model (i.e. with more parameters) is not reflected appropriately.

Figure 11 seems to show the wide band of uncertainty (but this is not clearly explained in the figure caption). Can the apparent difference/improvement of the model prediction be justified, given the considerable underlying uncertainty?

Some of the figures are still very hard to read. Axis labels and numbers are generally too small. Fig. 3, for example, is improved as far as resolution is concerned, but font size is far too small. The same applies to other figures.

Best regards, Jens-Arne Subke
* * *

---

## Author Response (AR1)

Dear Drs Zha and Zhuang,

Thank you for responding to the referee reports. Work on the manuscript has addressed many of the concerns by referees, and I am content with these (e.g. the issue of equifinality). However, I am not convinced that you have managed to alleviate the main point of referee 1, which is that the uncertainty associated with both models compared here is so large that it is not possible to judge any potential improvement from incorporating dormancy. I would therefor like to ask you to revise the manuscript further to address the issue of modelling uncertainty. I include some further specific explanations below.

**Response:** *We highly appreciate Dr. Subke's constructive comments. Below we detail how we have revised the manuscript by specifically further addressing the model uncertainty. We have also addressed some other minor comments regarding figure quality.*

I agree with referee 1 that model uncertainty is key for this comparison, and this is not well illustrated also in the revised paper. Figures 2, 4 and 5 show that in some instances the new model follows temporal dynamics better, but the improvement over the nondormancy model is not that clear, and in absence of explicit uncertainty associated with either model, the reader can not judge whether this is a significant improvement. This then leads to similar problems when making regional predictions – Are stated differences for the contrasting models within the model uncertainty?

**Response:** *We used Fig. 2 to show dormancy model performance in comparison with observational data. We used different sets of observational data to verify both dormancy and nodormancy models in Figs. 4 and 5. While we acknowledge both models have various temporal dynamics in comparison with observational data, especially considering relatively large uncertainty associated with dormancy model (also for nodormancy model). However, we do see in general the dormancy model performs better. To confirm this, we plotted the linear model comparison with observational data for both models shown in Fig. 6. The comparison shows that the dormancy model MIC-TEM-Dormancy performs better with smaller intercepts and larger slopes that are closer to 1. For the simulated NEP of both models, we compared them with observational data in Table 5. In general, the dormancy model has relatively smaller intercepts while slopes and R values varying for various ecosystems. With recognition of these site-level uncertainties, we conducted ensemble simulations in the last revision to quantify the regional uncertainties for both models. Our results were presented with uncertainties in Figs. 11 and 12. Consequently, our conclusions were drawn by considering these uncertainties. In this revision, we revised the Abstract and Conclusion sections to reflect these changes concerning model differences and uncertain parameters and simulations by adding standard deviations.*

You also don't address fully the point that the dormancy model increases modelling uncertainty owing to the larger number of parameters. As no modelling uncertainty is quantified at present, I think that this fundamental limitation of constructing a more complex model (i.e. with more parameters) is not reflected appropriately.
Figure 11 seems to show the wide band of uncertainty (but this is not clearly explained in the figure caption). Can the apparent difference/improvement of the model prediction be justified, given the considerable underlying uncertainty?

*Response:   Thanks for the comments.  We agree that adding the dormancy effects makes the model more complex by increasing number of parameters, which might increase model uncertainty.  This is a fundamental modeling dilemma, i.e., shall we use a simpler model (less processes considered) or a more adequate model (more essential processes considered).  In this study, we believe, the dormancy dynamics are an important process to be considered to more adequately capture microbial decomposition. Two versions of the model simulations and observational data comparisons for dominant ecosystems in the region indicate that the dormancy model better captures the observation while there are uncertainties for both models. We agree this study has not fully addressed this problem – which is subject to a research community debate.  Therefore, in this revision, we added a paragraph to discuss this modeling dilemma, quoted below "While our analysis suggests it is important to incorporate microbial dormancy dynamics into a process-based biogeochemistry model to more adequately simulate carbon dynamics in northern high latitudes, we do confront modeling dilemmas.  First, our process-based models have a relatively large number of parameters, which unavoidably creates the "equifinality" problem as recognized in our previous studies for the model (e.g., Tang and Zhuang, 2008, 2009).  To alleviate this problem in this analysis, we have conducted parameter ensemble simulations at both site and regional levels and presented our results with uncertainties, which could be a standard approach for process-based complex biogeochemistry modeling analyses.  Second, incorporating more ecosystem processes increases the number of parameters in our model, inducing even larger uncertainties for both site level and regional simulations.  On the one hand, the more complex model to a certain degree helps capture observations, on the other hand, the model uncertainty has not been constrained or even enlarged. We highlight the need to further investigate this trade-off within the modeling research community.".*

Some of the figures are still very hard to read. Axis labels and numbers are generally too small. Fig. 3, for example, is improved as far as resolution is concerned, but font size is far too small. The same applies to other figures.

*Response:  Thanks much for the suggestion.  In this revision, we improved the quality of these figures.*

Best regards, Jens-Arne Subke

[revised manuscript text omitted]

---

## Author Response (AR2)

**Associate Editor Decision: Reconsider after major revisions** (02 Jul 2020) by Jens-Arne Subke
Comments to the Author:
Dear Drs Zha and Zhuang,

Thank you for addressing referees concerns over the manuscript. I have reviewed your changes and have reason to request further changes before I'm able to finally accept this manuscript for publication. This addresses mostly the writing and framing of your science, not further analyses, and I hope that you will be able to perform revisions relatively swiftly.

A key concern is your framing of the study as addressing "Arctic" soils. Your study does not focus on Arctic ecosystems. Latitudes north of 45° include mostly temperate and boreal regions, with the Arctic making up a small fraction only (possibly 10 - 15%, depending on definition). Your reference sites (Table 2) include only two actual Arctic sites. It's not clear to me why you present this as a study addressing Arctic C storage; the model addresses microbial dormancy in general, why is it framed specifically around the Arctic context? Please remove or revise sections that present this study as specifically Arctic-focused (e.g. lines 48/49 and 58 in the Abstract, and lines 114-120, and 126/127). Similarly, "high latitudes" would normally be understood to be > 60°, most commonly in fact > 66° (i.e. pole-ward from Arctic Circle). Please remove reference to "high latitude" where you include areas north of 45°N.

**Response: Thanks for the concern. In this revision, we use "northern temperate and boreal" instead of "Arctic" or "High Latitudes" throughout the text. Consequently, the title is changed to "*Microbial dormancy and its impacts on northern temperate and boreal terrestrial ecosystem carbon budget*".**

This aside, I also think that you have to make it much clearer that this presents the potential of including microbial dormancy in models to potential outcomes, rather than providing realistic scaled numbers for northern regions. A very substantial part of land surface north of 45° is agricultural, which is entirely absent from your data sets used for parameterisation. Please make it absolutely clear that this is an exercise of using models to estimate potential regional C dynamics, but that this applies to natural ecosystems only, so it is a partial C balance for the region. I see no problem with this being a study to demonstrate the role of dormancy in principle, but want to avoid these numbers being used to claim actual source or sink functions of ecosystems in the region. Your objectives stated at the end of the introduction should make this very clear.

**Response: Thanks for the suggestion. In this revision, we made clearer that the study was focusing on natural ecosystems only and revised the text accordingly.**

Please note the detailed points below when you re-draft the manuscript. I hope that this will make the manuscript more focused and comprehensive.

141: Again, why the Arctic specifically? Your modeling approach is generic to any soil microbial dormancy, or not?

**Response: Made correction.**

166/184: Equations 1 and 3 are identical, but use different parameters. Please keep consistent (and possibly omit one of them, as they are repetitive).

**Response: Deleted Eqn. 1 and re-ordered equations.**

247-248: Please refer to biomes covered as "northern ecosystems ranging from temperate to Arctic", or similar. Not "high latitude".

**Response: Changed.**

260: This is misleading. Four of the validation sites are the same as those used for parameterisation. This has to be absolutely clear.

**Response:** Indeed a few sites are overlapped between Tables 2 and 3 to be able to cover all ecosystem types in this study for both parameterization and validation. However, for those overlapped sites, we used the data for different years for parameterization and validation. Specifically, the overlapped sites are Howland Forest, UCI-1964, Atqasuk, and Ivotuk. At Howland Forest site, we used the data during 01/2003-12/2003 for validation, the data during 01/2004-12/2004 for parameterization. At UCI-1964 site, the data during 01/2004-10/2005 are used for parameterization, but the data during 01/2002-12/2003 are for validation. At Atqasuk site, the data of 01/2003-12/2004 are for model validation, the data of 01/2005-12/2006 are for model parameterization. At Ivotuk site, the data during 01/2004-12/2004 are for parameterization, but the data of 01/2005-12/2005 are for validation. To make this clear, we added a sentence to clarify this in main text "*Four of these six sites were also used for parameterization (Table 2). However, we used the data of different observation periods for model validation for those overlapped sites.* "

337-339: This sentence goes beyond data presentation. Please move it to the discussion, where this interpretation is more appropriate.

**Response:** Deleted the sentence in these lines, but moved the citations to Discussion section to support the argument of soil decomposition, N availability and NPP differences.

– 360: Also here, avoid lengthy explanations in the results section. Abbreviate the entire section to: "Without considering dormancy, MIC-TEM estimates more active microbial biomass, hence overestimating both RH, and NPP (due to higher simulated N mineralization and uptake by plants), but resulting in lower NEP than that calculated by MIC-TEM-dormancy."

**Response:** Thanks for the suggestion. We used your more concise version in this revision.

361-373: This section is not very well phrased. Please present the key findings of the seasonal dynamics without detailed descriptions of the shape of graphs. Also no need to reference the fact that moisture and temperature are higher in summer.

**Response:** Following your suggestion, in this revision, we made the paragraph more concise to "*Temporally, both models projected higher NPP and RH in summer than in winter (Figures 8a and 8b) due to higher soil temperature and moisture (McGuire et al., 1992). Setting the RH projection from MIC-TEM as a baseline, MIC-TEM-dormancy projected 33% less RH in summer (May to September), and 30% more in winter (other months) (Figure 8b), indicating that without dormancy, model tends to estimate lower soil respiration due to ignorance of dormant respiration in winter, but higher soil respiration due to higher active biomass in summer. NEP seasonality estimated with two models are close to each other (Figure 8c), but the dormancy model projected slightly higher NEP in summer.*"

397-398: Delete last sentence (again, this should be picked up in the discussion).

**Response: We removed this sentence.**

399-400: Delete first sentence, and add "in the 2090s" after "scenarios" in the following sentence. Note that it is 2090s, not "2990s" as stated in line 399.

**Response: Thanks. We corrected the error.**

410, 413 and 415: Replace "predicted" by "predicts".

**Response: Changed.**

– 426: This should be deleted as it has been covered in the introduction. Focus on key findings of your study in the discussion.

**Response: Following your suggestion, we deleted these sentences.**

Figure 3: The figure caption states that you show parameter ranges for all six sites, but on three are shown. Text in the Results section also suggests that data are shown for all six sites.

**Response:  We made these consistent in this revision.**

[revised manuscript text omitted]